# The Randomized Midpoint Method for Log-Concave Sampling

**Ruoqi Shen**
University of Washington
shenr3@cs.washington.edu

**Yin Tat Lee**
University of Washington and Microsoft Research
yintat@uw.edu

## Abstract

Sampling from log-concave distributions is a well researched problem that has many applications in statistics and machine learning. We study the distributions of the form $p^* \propto \exp(-f(x))$, where $f : \mathbb{R}^d \to \mathbb{R}$ has an $L$-Lipschitz gradient and is $m$-strongly convex. In our paper, we propose a Markov chain Monte Carlo (MCMC) algorithm based on the underdamped Langevin diffusion (ULD). It can achieve $\epsilon \cdot D$ error (in 2-Wasserstein distance) in $\tilde{O}\left(\kappa^{7/6}/\epsilon^{1/3} + \kappa/\epsilon^{2/3}\right)$ steps, where $D \stackrel{\text{def}}{=} \sqrt{\frac{d}{m}}$ is the effective diameter of the problem and $\kappa \stackrel{\text{def}}{=} \frac{L}{m}$ is the condition number. Our algorithm performs significantly faster than the previously best known algorithm for solving this problem, which requires $\tilde{O}\left(\kappa^{1.5}/\epsilon\right)$ steps [7, 15]. Moreover, our algorithm can be easily parallelized to require only $O(\kappa \log \frac{1}{\epsilon})$ parallel steps.

To solve the sampling problem, we propose a new framework to discretize stochastic differential equations. We apply this framework to discretize and simulate ULD, which converges to the target distribution $p^*$. The framework can be used to solve not only the log-concave sampling problem, but any problem that involves simulating (stochastic) differential equations.

## 1 Introduction

In this paper, we study the problem of sampling from a high-dimensional log-concave distribution. This problem is central in statistics, machine learning and theoretical computer science, with applications such as Bayesian estimation [1], volume computation [55] and bandit optimization [54]. In a seminal 1989 result, Dyer, Frieze and Kannan [23] first presented a polynomial-time algorithm (for an equivalent problem) that takes $\tilde{O}(d^{23} \log \frac{1}{\epsilon})$ steps on any $d$ dimensional log-concave distribution to achieve target accuracy $\epsilon$. After three decades of research in Markov chain Monte Carlo (MCMC) and convex geometry [34, 2, 22, 35, 27, 37, 11, 30, 31, 43], results have been improved to $\tilde{O}(d^4 \log \frac{1}{\epsilon})$ steps for general log-concave distributions and slightly better for distributions given in a certain form. Unfortunately, $d \log \frac{1}{\epsilon}$ steps are necessary even for a special case of log-concave sampling, i.e., convex optimization [3]. To avoid this lower bound, there has been a recent surge of interest in obtaining a faster algorithm via assuming some properties on the distribution.

We call a distribution *log-concave* if its density is proportional to $e^{-f(x)}$ with a convex function $f$. For the standard assumption that $f$ is $m$-strongly convex with an $L$-Lipschitz gradient (see Section 3.1), the current best algorithms have at least a linear $d$ or $1/\epsilon$ dependence or a large dependence on the condition number $\kappa \stackrel{\text{def}}{=} \frac{L}{m}$. In this paper, we present an algorithm with no dependence on $d$ and a much smaller dependence on $\kappa$ and $\epsilon$ than shown in previous research. Moreover, our algorithm is the first algorithm with better than $1/\epsilon$ dependence that is not Metropolis-adjusted and does not make any extra assumption, such as high-order smoothness [41, 42, 6, 45].

To explain our main result, we note that this problem has an effective diameter $D \stackrel{\text{def}}{=} \sqrt{\frac{d}{m}}$ because the distance between the minimizer $x^*$ of $f$ and a random point $y \sim e^{-f}$ satisfies $\mathbb{E}_{y \sim e^{-f}} \|x^* - y\|^2 \leq \frac{d}{m}$[19]. Therefore, a natural problem definition[1] is to find a random $x$ that makes the Wasserstein distance small:

$$W_2(x, y) \leq \epsilon \cdot D. \tag{1}$$

This choice of distance is also common in previous papers [19, 20, 10, 41, 29, 42, 6].

For $\epsilon = 1$, we can simply output the minimizer $x^*$ of $f$ as the "random" point. We first consider the question how quickly we can find a random point satisfying $\epsilon = \frac{1}{2}$. For convex optimization under the same assumption, it takes $\sqrt{\kappa}$ iterations via acceleration methods or $d$ iterations via cutting plane methods, and these results are tight. For sampling, the current fastest algorithms take either $\tilde{O}(\kappa^{1.5})$ steps [7, 15] or $\tilde{O}(d^4)$ steps [36]. Although there is no rigorous lower bound for this problem, it is believed that $\min(\kappa, d^2)$ is the natural barrier.[2] This paper presents an algorithm that takes only $\tilde{O}(\kappa^{7/6})$ steps, much closer to the natural barrier of $\kappa$ for the high-dimensional regime.

For general $0 < \epsilon < 1$, our algorithm takes $\tilde{O}(\kappa^{7/6}/\epsilon^{1/3} + \kappa/\epsilon^{2/3})$ steps, which is almost linear in $\kappa$ and sub-linear in $\epsilon$. It has significantly better dependence on both $\kappa$ and $\epsilon$ than previous algorithms. (See the detailed comparison in Table 1.) Moreover, if we query gradient $\nabla f$ at multiple points in parallel in each step, we can improve the number to $O(\kappa \log \frac{1}{\epsilon})$ steps.

## 1.1 Contributions

We propose a new framework to discretize stochastic differential equations (SDEs), which is a crucial step of log-sampling algorithms. Since our techniques can also be applied to ordinary differential equations (ODEs), we focus on the following ODE here:

$$\frac{\mathrm{d}x}{\mathrm{d}t} = F(x(t)).$$

There are two main frameworks to discretize a differential equation. One is the Taylor expansion, which approximates $x(t)$ by $x(0) + x'(0)t + x''(0)\frac{t^2}{2} + \cdots$. Our paper uses the second framework, called the *collocation method*. This method uses the fact that the differential equation is equivalent to the integral equation $x = \mathcal{T}(x)$, where $\mathcal{T}$ maps continuous functions to continuous functions:

$$\mathcal{T}(x)(t) = x(0) + \int_0^t F(x(s)) \, \mathrm{d}s \text{ for all } t \geq 0.$$

Since $x$ is a fixed point of $\mathcal{T}$, we can approximate $x$ by computing $\mathcal{T}(\mathcal{T}(\cdots(\mathcal{T}(x_0))\cdots))$ for some approximate initial function $x_0$. Algorithmically, two key questions are how to: (1) show when and how quickly $\mathcal{T}$ iterations converge, and (2) compute the integration. The convergence rate of $\mathcal{T}$ was shown by the Picard–Lindelöf Theorem in the 1890s [32, 48] and was key to achieving $O(\kappa^{1.75})$ and $O(\kappa^{1.5})$ in the previous papers [29, 7]. To approximate the integration, one standard approach is to approximate

$$\int_0^t F(x(s)) \, \mathrm{d}s \sim \sum_i w_i F(x(s_i))$$

| | # Step | |
| :---: | :---: | :---: |
| **Algorithm** | **Warm Start** | **Cold Start** |
| Hit-and-Run[36] | $\tilde{O}\left(d^3 \log(\frac{1}{\epsilon})\right)$ | $\tilde{O}\left(d^4 \log(\frac{1}{\epsilon})\right)$ |
| Langevin Diffusion[19, 13] | $\tilde{O}\left(\kappa^2/\epsilon^2\right)$ | |
| Underdamped Langevin Diffusion [10] | $\tilde{O}\left(\kappa^2/\epsilon\right)$ | |
| Underdamped Langevin Diffusion2 [15] | $\tilde{O}\left(\kappa^{1.5}/\epsilon + \kappa^2\right)$ | |
| High-Order Langevin Diffusion[45] | $\tilde{O}\left(\kappa^{19/4}/\epsilon^{1/2} + \kappa^{13/3}/\epsilon^{2/3}\right)$ | |
| Metropolis-Adjusted Langevin Algorithm[21] | $\tilde{O}\left(\left(\kappa d + \kappa^{1.5}\sqrt{d}\right)\log(\frac{1}{\epsilon})\right)$ | $\tilde{O}\left(\left(\kappa d^2 + \kappa^{1.5}d^{1.5}\right)\log(\frac{1}{\epsilon})\right)$ |
| Hamiltonian Monte Carlo with Euler Method [41] | $\tilde{O}\left(\kappa^{6.5}/\epsilon\right)$ | |
| Hamiltonian Monte Carlo with Collocation Method [29] | $\tilde{O}\left(\kappa^{1.75}/\epsilon\right)$ | |
| Hamiltonian Monte Carlo with Collocation Method 2 [7] | $\tilde{O}\left(\kappa^{1.5}/\epsilon\right)$ | |
| Underdamped Langevin Diffusion with Randomized Midpoint Method (This Paper) | $\tilde{O}\left(\kappa^{7/6}/\epsilon^{1/3} + \kappa/\epsilon^{2/3}\right)$ | |

Table 1: Summary of iteration complexity. Except for Hit-and-Run, each step involves $O(1)$-gradient computation. Hit-and-Run takes $\tilde{O}(1)$ function value computations in each step. [15, 45] assume the hessian of $f$ is Lipschitz, which is stronger than our assumption.

for some carefully chosen $w_i$ and $s_i$. The key drawback of this approach is its introduction of a deterministic error, which accumulates linearly to the number of steps. Since we expect to take at least $\kappa$-many iterations, the approximation error must be $\kappa$ times smaller than the target accuracy.

In this paper, we improve upon the collocation method for sampling by developing a new algorithm, called the *randomized midpoint method*, that yields three distinct benefits:

1. We generalize fixed point iteration to stochastic differential equations and hence avoid the cost of reducing SDEs to ODEs, as was done in [29].

2. We greatly reduce the error accumulation by simply approximating $\int_0^t F(x(s))ds$ by $t \cdot F(x(s))$ where $s$ is randomly chosen from 0 to $t$ uniformly.

3. We show that two iterations of $\mathcal{T}$ suffice to achieve the best theoretical guarantee.

Although we discuss only strongly convex functions with a Lipschitz gradient, we believe our framework can be applied to other classes of functions, as well. By designing suitable unbiased estimators of integrals, researchers can easily use our approach to obtain faster algorithms for solving SDEs that are unrelated to sampling problems.

## 1.2 Paper Organization

Section 2 provides background information on solving the log-concave sampling problem, while Section 3 introduces our notations and assumptions about the function $f$. We introduce our algorithm in Section 4, where we present the main result of our paper. We show our proofs in appendices: Appendix A–how we simulate the Brownian motion; Appendix B–important properties of ULD and the Brownian motion; Appendix C– bounds for the discretization error of our algorithm; Appendix D–a bound on the average value of $\|\nabla f(x_n)\|$ and $\|v_n\|$ in our algorithm, which is useful for bounding the discretization error; Appendix E–proofs for the main result of our paper; Appendix F–additional proofs on how to parallelize our algorithm.

## 2  Background

Many different algorithms have been proposed to solve the log-concave sampling problem. The general approach uses a MCMC-based algorithm that often includes two steps. The first step involves the choice of a Markov process with a stationary distribution equal or close to the target distribution. The second step is discretizing the process and simulating it until the distribution of the points generated is sufficiently close to the target distribution.

### 2.1  Choosing the Markov Process

One commonly used Markov process is the Langevin diffusion (LD) [52, 25, 18]. LD evolves according to the SDE

$$\mathrm{d}x(t) \;=\; -\nabla f(x(t))\,\mathrm{d}t + \sqrt{2}\,\mathrm{d}B_t, \tag{2}$$

where $B_t$ is the standard Brownian motion. Under the assumption that $f$ is $L$-smooth and $m$-strongly convex (see Section 3.1) with $\kappa = \frac{L}{m}$ as the condition number, [19, 13, 8] show that algorithms based on LD can achieve less than $\epsilon$ error in $\tilde{O}\left(\frac{\kappa^2}{\epsilon^2}\right)$ steps. Other related works include LD with stochastic gradient [14, 57, 50, 6] and LD in the non-convex setting [50, 9].

One important breakthrough introduced the Hamiltonian Monte Carlo (HMC), originally proposed in [28]. In this process, SDE (2) is approximated by a piece-wise curve, where each piece is governed by an ODE called the Hamiltonian dynamics. The Hamiltonian dynamics maintains a velocity $v$ in addition to a position $x$ and conserves the value of the Hamiltonian $H(x,v) = f(x) + \frac{1}{2}\left\|v\right\|^2$. HMC has been widely studied in [46, 40, 41, 42, 29, 7, 31]. The works [7, 15] show that algorithms based on HMC can achieve less than $\epsilon$ error in $\tilde{O}\left(\frac{\kappa^{1.5}}{\epsilon}\right)$ steps.

The underdamped Langevin diffusion (ULD) can be viewed as a version of HMC that replaces multiple ODEs with one SDE; it has been studied in [10, 24, 15]. ULD follows the SDE:

$$\mathrm{d}v(t) = -2v(t)\,\mathrm{d}t - u\nabla f(x(t))\,\mathrm{d}t + 2\sqrt{u}\,\mathrm{d}B_t, \qquad \mathrm{d}x(t) = v(t)\,\mathrm{d}t, \tag{3}$$

where $u = \frac{1}{L}$. [10] shows that even a basic discretization of ULD has a fast convergence rate that can achieve less than $\epsilon$ error in $\tilde{O}\left(\frac{\kappa^2}{\epsilon}\right)$ steps. Recently, it was shown that ULD can be viewed as an accelerated gradient descent for sampling [39]. This suggests that ULD might be one of the right dynamic for sampling in the same way as the accelerated gradient descent method is appropriate for convex optimization. For this reason, our paper focuses on how to discretize ULD. We note that our framework can be applied to both LD and HMC to improve on previous results for these dynamics as well.

### 2.2  Discretizing the Process

To simulate the random process mentioned, previous works usually apply the Euler method [10, 19] or the Leapfrog method [41, 42] to discretize the SDEs or the ODEs. In Section 4.2, we introduce a 2-step fixed point iteration method to solve general differential equations. We apply this method to ULD and significantly reduce the discretization error compared to existing methods. In particular, ULD can achieve less than $\epsilon$ error in $\tilde{O}\left(\frac{\kappa^{7/6}}{\epsilon^{1/3}} + \frac{\kappa}{\epsilon^{2/3}}\right)$ steps. Table 1 summarizes the number of steps needed by previous algorithms versus our algorithm. Moreover, with slightly more effort, our algorithm can be parallelized so that it needs only $O\left(\kappa \log \frac{1}{\epsilon}\right)$ parallel steps.

On top of the discretization method, one can use a Metropolis-Hastings accept-reject step to ensure that the post-discretization random process results in a stationary distribution equal to the target distribution [4, 35, 53, 44, 33, 36, 38]. [36] gives the current best algorithm for arbitrary log-concave distribution. Originally proposed in [52, 53], the Metropolis Adjusted Langevin Algorithm (MALA) [51, 26, 49, 5, 56, 47] applies the Metropolis-Hastings accept-reject step to the Langevin diffusion. [21] shows MALA can achieve $\epsilon$ error in total variation distance in $\tilde{O}\left(\left(\kappa d + \kappa^{1.5}\sqrt{d}\right)\log\left(\frac{\beta}{\epsilon}\right)\right)$ steps for $\beta$-warm start. Unlike other algorithms that have a $\frac{1}{\epsilon^{O(1)}}$ dependence on $\epsilon$, MALA depends logarithmically on $\epsilon$. However, $\beta$ usually depends exponentially on the dimension $d$, which results

in a $\Omega(d^{1.5})$ dependence in total. Since this paper focuses on achieving a dimension independent result, we do not discuss how to combine our process with a Metropolis-Hastings step in this paper.

Finally, we note that all results–including ours–can be improved if we assume that $f$ has bounded higher-order derivatives. To ensure a fair comparison in Table 1, we only include results that only assume $f$ is strongly convex and has a Lipschitz gradient.

# 3   Notations and Definitions

For any function $f$, we use $\tilde{O}(f)$ to denote the class $O(f) \cdot \log^{O(1)}(f)$. For vector $v \in \mathbb{R}^d$, we use $\|v\|$ to denote the Euclidean norm of $v$.

## 3.1   Assumptions on $f$

We assume that the function $f$ is a twice continuously differentiable function from $\mathbb{R}^d$ to $\mathbb{R}$ that has an $L$-Lipschitz continuous gradient and is $m$-strongly convex. That is, there exist positive constants $L$ and $m$ such that for all $x, y \in \mathbb{R}^d$,

$$\|\nabla f(x) - \nabla f(y)\| \le L \|x - y\|, \text{ and } f(y) \ge f(x) + \langle \nabla f(x), y - x \rangle + \frac{m}{2} \|x - y\|^2.$$

It is easy to show that these inequalities are equivalent to $mI_d \preceq \nabla^2 f(x) \preceq LI_d$, where $I_d$ is the identity matrix of dimension $d$. Let $\kappa = \frac{L}{m}$ be the condition number. We assume that we have access to an oracle that, given a point $x \in \mathbb{R}^d$, can return the gradient of $f$ at point $x$, $\nabla f(x)$.

## 3.2   Wasserstein Distance

The $p$th Wasserstein distance between two probability measures $\mu$ and $\nu$ is defined as

$$W_p(\mu, \nu) \;\; = \;\; \left( \inf_{(X,Y) \in \mathcal{C}(\mu,\nu)} \mathbb{E}\left[ \|X - Y\|^p \right] \right)^{1/p},$$

where $\mathcal{C}(\mu, \nu)$ is the set of all couplings of $\mu$ and $\nu$. In this paper, for any $0 < \epsilon < 1$, we study the number of steps needed so that the $W_2$ distance between the distribution of the point our algorithms generate and the target distribution is smaller than $\epsilon \cdot D$.

# 4   Algorithms and Results

## 4.1   Underdamped Langevin Diffusion (ULD)

ULD is a random process that evolves according to (3). Our paper studies (3) with $u = \frac{1}{L}$. Under mild conditions, it can be shown that the stationary distribution of (3) is proportional to $\exp\left(-f(x) + L\|v\|^2/2\right)$. Then, the marginal distribution of $x$ is proportional to $\exp(-f(x))$. It can also be shown that the solution to (3) has a contraction property [10, 24], shown in the following lemma.

**Lemma 1** (Theorem 5 of [10]). *Let $(x_0, v_0)$ and $(y_0, w_0)$ be two arbitrary points in $\mathbb{R}^d \times \mathbb{R}^d$. Let $(x_t, v_t)$ and $(y_t, w_t)$ be the exact solutions of the underdamped Langevin diffusion after time $t$. If $(x_t, v_t)$ and $(y_t, w_t)$ are coupled through a shared Brownian motion, then,*

$$\mathbb{E}\left[ \|x_t - y_t\|^2 + \|(x_t + v_t) - (y_t + w_t)\|^2 \right] \le e^{-\frac{t}{\kappa}} \mathbb{E}\left[ \|x_0 - y_0\|^2 + \|(x_0 + v_0) - (y_0 + w_0)\|^2 \right].$$

This contraction bound can be very useful for showing the convergence of the continuous process (3). In our algorithm, we discretize the continuous process to implement it; therefore we need to use this contraction bound together with a discretization error bound to show the guarantee of our algorithm. In Section 4.2, we show how we discretize (3).

---

**Algorithm 1** Randomized Midpoint Method for ULD

---

1: **Procedure** RandomMidpoint($x_0, v_0, N, h$)
2: **For** $n = 0, ..., N - 1$
3:    Randomly sample $\alpha$ uniformly from $[0, 1]$.
4:    Generate Gaussian random variable $\left( W_1^{(n)}, W_2^{(n)}, W_3^{(n)} \right) \in \mathbb{R}^{3d}$ as in Appendix A
5:    $x_{n+\frac{1}{2}} = x_n + \frac{1}{2} \left( 1 - e^{-2\alpha h} \right) v_n - \frac{1}{2} u \left( \alpha h - \frac{1}{2}(1 - e^{-2\alpha h}) \right) \nabla f(x_n) + \sqrt{u} W_1^{(n)}$.
6:    $x_{n+1} = x_n + \frac{1}{2} \left( 1 - e^{-2h} \right) v_n - \frac{1}{2} u h \left( 1 - e^{-2(h-\alpha h)} \right) \nabla f(x_{n+\frac{1}{2}}) + \sqrt{u} W_2^{(n)}$.
7:    $v_{n+1} = v_n e^{-2h} - u h e^{-2(h-\alpha h)} \nabla f(x_{n+\frac{1}{2}}) + 2\sqrt{u} W_3^{(n)}$.
8: **end for**
9: **end procedure**

---

## 4.2 Randomized Midpoint Method

Our step size for each iteration is $h$. In iteration $n$ of our algorithm, to simulate (3), we need to approximate the solution to SDE (3) at time $h$, $(x_n^*(h), v_n^*(h))$, with initial value, $(x_n, v_n)$. The simplest way to do so is to use the Euler method:

$$v_n(h) = (1 - 2h)v_n - uh\nabla f(x_n) + 2\sqrt{uh}\zeta, \quad x_n(h) = x_n + hv_n,$$

where $\zeta \in \mathbb{R}^d$ is drawn from the standard normal distribution. This discretization was considered in [20, 13] due to its simplicity.

As discussed in Section 1.1, we improve the accuracy by studying the integral formulation of (3):

$$x_n^*(t) = x_n + \frac{1 - e^{-2t}}{2}v_n - \frac{u}{2}\int_0^t \left( 1 - e^{-2(t-s)} \right) \nabla f(x_n^*(s)) \, \mathrm{d}s + \sqrt{u}\int_0^t \left( 1 - e^{-2(t-s)} \right) \mathrm{d}B_s,$$

$$v_n^*(t) = v_n e^{-2t} - u \left( \int_0^t e^{-2(t-s)}\nabla f(x_n^*(s)) \, \mathrm{d}s \right) + 2\sqrt{u}\int_0^t e^{-2(t-s)} \, \mathrm{d}B_s. \tag{4}$$

[10] considered the same integral formulation and used $\nabla f(x_n)$ to approximate $\nabla f(x_n^*(t))$ for $t \in [0, h]$ to get the following algorithm:

$$\hat{x}_n(h) = x_n + \frac{1 - e^{-2h}}{2}v_n - \frac{u}{2}\int_0^h \left( 1 - e^{-2(h-s)} \right) \nabla f(x_n) \, \mathrm{d}s + \sqrt{u}\int_0^h \left( 1 - e^{-2(h-s)} \right) \mathrm{d}B_s,$$

$$\hat{v}_n(h) = v_n e^{-2h} - u \left( \int_0^h e^{-2(h-s)}\nabla f(x_n) \, \mathrm{d}s \right) + 2\sqrt{u}\int_0^h e^{-2(h-s)} \, \mathrm{d}B_s.$$

However, this approximation method can still generate a relatively large error. Our paper proposes a new method, the randomized midpoint method, to solve (4), which yields a more accurate approximation and significantly reduces the total runtime of the algorithm.

We first need to identify an accurate estimator of the integral $\int_0^h \left( 1 - e^{-2(h-s)} \right) \nabla f(x_n^*(s)) \, \mathrm{d}s$. To do so, we sample a random number $\alpha$ uniformly from $[0, 1]$ so that $\alpha h$ gives a random point from $[0, h]$. Then, $h \left( 1 - e^{-2(h-\alpha h)} \right) \nabla f(x_n^*(\alpha h))$ is an accurate estimator of the integral $\int_0^h \left( 1 - e^{-2(h-s)} \right) \nabla f(x_n^*(s)) \, \mathrm{d}s$. We can further show that this estimator is unbiased.

For brevity, we use $x_{n+\frac{1}{2}}$ to denote our approximation of $x_n^*(\alpha h)$. To approximate $x_n^*(\alpha h)$, we use equation (4) again:

$$x_{n+\frac{1}{2}} = x_n + \frac{1 - e^{-2\alpha h}}{2}v_n - \frac{u}{2}\int_0^{\alpha h} \left( 1 - e^{-2(\alpha h-s)} \right) \nabla f(x_n) \mathrm{d}s + \sqrt{u}\int_0^{\alpha h} \left( 1 - e^{-2(\alpha h-s)} \right) \mathrm{d}B_s.$$

Then, $(x_n^*(h), v_n^*(h))$ can be approximated as

$$x_{n+1} = x_n + \frac{1 - e^{-2h}}{2}v_n - \frac{u}{2}h \left( 1 - e^{-2(h-\alpha h)} \right) \nabla f(x_{n+\frac{1}{2}}) + \sqrt{u}\int_0^h \left( 1 - e^{-2(h-s)} \right) \mathrm{d}B_s,$$

$$v_{n+1} = v_n e^{-2h} - uhe^{-2(h-\alpha h)}\nabla f(x_{n+\frac{1}{2}}) + 2\sqrt{u}\int_0^h e^{-2(h-s)}\,\mathrm{d}B_s.$$

Note that we can view (4) as the fixed point of the operator $\mathcal{T}$, $x_n^* = \mathcal{T}(x_n^*)$, where for all $t$,

$$\mathcal{T}(x)(t) = x_n + \frac{1-e^{-2t}}{2}v_n - \frac{u}{2}\int_0^t \left(1 - e^{-2(t-s)}\right)\nabla f(x(s))\,\mathrm{d}s + \sqrt{u}\int_0^t \left(1 - e^{-2(t-s)}\right)\mathrm{d}B_s. \tag{5}$$

Then, our randomized algorithm is essentially approximating $\mathcal{T}(\mathcal{T}(x_n))$. Under the assumption $f$ is twice differentiable, we show that two iterations suffice to achieve the best theoretical guarantee, but we suspect more iterations might be useful if $f$ has higher order derivatives. As emphasized in Section 1.1, the way we obtain our algorithm forms a general framework that can be applied to other SDEs.

In Lemma 5, we show that the stochastic terms $W_1 = \int_0^{\alpha h}\left(1 - e^{-2(\alpha h - s)}\right)\mathrm{d}B_s$, $W_2 = \int_0^h \left(1 - e^{-2(h-s)}\right)\mathrm{d}B_s$, and $W_3 = \int_0^h e^{-2(h-s)}\,\mathrm{d}B_s$ conditional on the choice of $\alpha$ follow a multi-dimensional Gaussian distribution and therefore can be easily sampled. The steps mentioned above are summarized in Algorithm 1. Using this randomized midpoint method, we can solve (4) much more accurately than previous works. We show that the discretization error satisfies:

**Lemma 2.** *For each iteration $n$ of Algorithm 1, let $\mathbb{E}_\alpha$ be the expectation taken over the random choice of $\alpha$ in iteration $n$. Let $\mathbb{E}$ be the expectation taken over other randomness in iteration $n$. Let $(x_n^*(t), v_n^*(t))_{t\in[0,h]}$ be the solution of the exact underdamped Langevin diffusion starting from $(x_n, v_n)$ coupled through a shared Brownian motion with $x_{n+\frac{1}{2}}$, $v_n$ and $x_{n+1}$. Assume that $h \leq \frac{1}{20}$ and $u = \frac{1}{L}$. Then, $x_{n+1}$ and $v_{n+1}$ of Algorithm 1 satisfy*

$$\mathbb{E}\left\|\mathbb{E}_\alpha x_{n+1} - x_n^*(h)\right\|^2 \leq O\left(h^{10}\left\|v_n\right\|^2 + u^2 h^{12}\left\|\nabla f(x_n)\right\|^2 + udh^{11}\right),$$

$$\mathbb{E}\left\|x_{n+1} - x_n^*(h)\right\|^2 \leq O\left(h^6\left\|v_n\right\|^2 + u^2 h^4\left\|\nabla f(x_n)\right\|^2 + udh^7\right),$$

$$\mathbb{E}\left\|\mathbb{E}_\alpha v_{n+1} - v_n^*(h)\right\|^2 \leq O\left(h^8\left\|v_n\right\|^2 + u^2 h^{10}\left\|\nabla f(x_n)\right\|^2 + udh^9\right),$$

$$\mathbb{E}\left\|v_{n+1} - v_n^*(h)\right\|^2 \leq O\left(h^4\left\|v_n\right\|^2 + u^2 h^4\left\|\nabla f(x_n)\right\|^2 + udh^5\right).$$

In Appendix D, we show that the average value of $\|v_n\|^2$ is of order $\tilde{O}\left(\frac{d}{L}\right)$; that of $\|\nabla f(x_n)\|^2$ is of order $\tilde{O}(Ld)$. Then, Lemma 2 shows that the bias of the discretization is of order $\tilde{O}\left(h^4\sqrt{\frac{d}{L}}\right)$ and the standard deviation is of order $\tilde{O}\left(h^2\sqrt{\frac{d}{L}}\right)$, which implies the error is larger when $h$ is larger. However, by Lemma 1, in order for the algorithm to converge in a small number of steps, we need to avoid choosing an $h$ that is too small. Therefore, it is important to choose the largest possible $h$ that can still make the algorithm converge. By Lemma 1, it is sufficient to run our algorithm for $\tilde{O}\left(\frac{\kappa}{h}\right)$ iterations. Then, the bias will cumulate to $\tilde{O}\left(h^4\sqrt{\frac{d}{L}}\cdot\frac{\kappa}{h}\right) = \tilde{O}\left(h^3\sqrt{\frac{d\kappa}{m}}\right)$, and the standard deviation will cumulate to $\tilde{O}\left(h^2\sqrt{\frac{d}{L}}\cdot\sqrt{\frac{\kappa}{h}}\right) = \tilde{O}\left(h^{1.5}\sqrt{\frac{d}{m}}\right)$. Thus, in order to make the $W_2$ distance less than $\tilde{O}\left(\epsilon\sqrt{\frac{d}{m}}\right)$, we show in Theorem 3 that it is enough to choose $h$ to be $\tilde{\Theta}\left(\min\left(\frac{\epsilon^{1/3}}{\kappa^{1/6}}, \epsilon^{2/3}\right)\right)$. This choice of $h$ yields the main result of our paper, which is stated in Theorem 3. (See Appendix E for the full proof.)

**Theorem 3** (Main Result). *Let $f$ be a function such that $0 \prec m\cdot I_d \preceq \nabla^2 f(x) \preceq L\cdot I_d$ for all $x \in \mathbb{R}^d$. Let $Y$ be a random point drawn from the density proportional to $e^{-f}$. Let the starting point $x_0$ be the point that minimizes $f(x)$ and $v_0 = 0$. For any $0 < \epsilon < 1$, if we set the step size of Algorithm 1 as $h = C\min\left(\frac{\epsilon^{1/3}}{\kappa^{1/6}}\log^{-1/6}\left(\frac{1}{\epsilon}\right), \epsilon^{2/3}\log^{-1/3}\left(\frac{1}{\epsilon}\right)\right)$, for some small constant $C$ and run the algorithm for $N = \frac{2\kappa}{h}\log\left(\frac{20}{\epsilon^2}\right) \leq \tilde{O}\left(\frac{\kappa^{7/6}}{\epsilon^{1/3}} + \frac{\kappa}{\epsilon^{2/3}}\right)$ iterations, then Algorithm 1 after*

---

**Algorithm 2** Randomized Midpoint Method for ULD (Parallel)

---

1: **Procedure** RandomMidpoint_P$(x_0, v_0, N, h, R)$
2: **For** $n = 0, ..., N-1$
3:     Randomly sample $\alpha_1, ..., \alpha_R$ uniformly from $\left[0, \frac{1}{R}\right], \left[\frac{1}{R}, \frac{2}{R}\right], ..., \left[\frac{R-1}{R}, 1\right]$.
4:     Generate Gaussian r.v. $\left(W_{1,1}^{(n)}, ..., W_{1,R}^{(n)}, W_2^{(n)}, W_3^{(n)}\right) \in \mathbb{R}^{(R+2)d}$ similar to Appendix A
5:     $x_n^{(0,i)} = x_n$ for $i = 1, ..., R$.
6:     **For** $k = 1, ..., K-1, i = 1, ..., R$
7:        $x_n^{(k,i)} = x_n + \frac{1}{2}\left(1 - e^{-2\alpha_i h}\right)v_n$
8:        $-\frac{1}{2}u\sum_{j=1}^{i}\left[\int_{(j-1)\delta}^{\min(j\delta, \alpha_i h)}\left(1 - e^{-2(\alpha_i h - s)}\right)\mathrm{d}s \cdot \nabla f(x_n^{(k-1,j)})\right] + \sqrt{u}W_{1,i}^{(n)}$
9:     **end for**
10:     $x_{n+1} = x_n + \frac{1}{2}\left(1 - e^{-2h}\right)v_n - \frac{1}{2}u\sum_{i=1}^{R}\delta\left(1 - e^{-2(h-\alpha_i h)}\right)\nabla f(x_n^{(K-1,i)}) + \sqrt{u}W_2^{(n)}$,
11:     $v_{n+1} = v_n e^{-2h} - u\sum_{i=1}^{R}\delta e^{-2(h-\alpha_i h)}\nabla f(x_n^{(K-1,i)}) + 2\sqrt{u}W_3^{(n)}$.
12: **end for**
13: **end procedure**

---

$N$ iterations can generate a random point $X$ such that $W_2(X, Y) \leq \epsilon\sqrt{\frac{d}{m}}$. Furthermore, each iteration of Algorithm 1 involves computing $\nabla f$ exactly twice.

### 4.3 A More General Algorithm

Now we show how our algorithm can be parallelized. The algorithm studied in this section can be viewed as a more general version of Algorithm 1. Instead of choosing one random point from $[0, h]$, we divide the time interval $[0, h]$ into $R$ pieces, each of length $\delta = \frac{h}{R}$, and choose one random point from each piece. That is, we randomly choose $\alpha_1, \alpha_2, ..., \alpha_R$ uniformly from $\left[0, \frac{1}{R}\right], \left[\frac{1}{R}, \frac{2}{R}\right], ..., \left[\frac{R-1}{R}, 1\right]$. As in Algorithm 1, to approximate $(x_n^*(h), v_n^*(h))$, we use

$$\tilde{x} = x_n + \frac{1 - e^{-2h}}{2}v_n - \frac{u}{2}\sum_{i=1}^{R}\delta\left(1 - e^{-2(h-\alpha_i h)}\right)\nabla f(x_n^*(\alpha_i h)) + \sqrt{u}\int_0^h\left(1 - e^{-2(h-s)}\right)\mathrm{d}B_s,$$

$$\tilde{v} = v_n e^{-2h} - u\sum_{i=1}^{R}\delta e^{-2(h-\alpha_i h)}\nabla f(x^*(\alpha_i h)) + 2\sqrt{u}\int_0^h e^{-2(h-s)}\mathrm{d}B_s,$$

which gives an unbiased estimator of $(x_n^*(h), v_n^*(h))$. The next step is to approximate $x_n^*(\alpha_i h)$ for $i = 1, .., R$. We know that the solution $x_n^*$ is the fixed point of the operator $\mathcal{T}$ defined in (5). To solve the fixed point of $\mathcal{T}$, we can use the fixed point iteration method, which applies the operator $\mathcal{T}$ multiple times on some initial point. By the Banach fixed point theorem, the resulting points can converge to the fixed point of $\mathcal{T}$. Instead of applying $\mathcal{T}$, which involves computing an integral, we apply the operator $\tilde{\mathcal{T}}$, which approximates $\mathcal{T}$, on $X = \left(x^{(1)}, ..., x^{(R)}\right)$,

$$\tilde{\mathcal{T}}(X)_i = x_n + \frac{1}{2}\left(1 - e^{-2\alpha_i h}\right)v_n - \frac{1}{2}u\sum_{j=1}^{i}\left[\int_{(j-1)\delta}^{\min(j\delta, \alpha_i h)}\left(1 - e^{-2(\alpha_i h - s)}\right)\mathrm{d}s \cdot \nabla f(x^{(j)})\right]$$

$$+ \sqrt{u}\int_0^{\alpha_i h}\left(1 - e^{-2(\alpha_i h - s)}\right)\mathrm{d}B_s.$$

We set the initial points to $x_n^{(0,j)} = x_n$ for $j = 1, ..., R$. Then, we apply $\tilde{\mathcal{T}}$ for $K$ times and get $(x^{(K,1)}, ..., x^{(K,R)}) = \tilde{\mathcal{T}}^{\circ K}(x^{(0,1)}, ..., x^{(0,R)})$. The preceding steps are summarized in Algorithm 2. It is easy to see Algorithm 1 is a special case of Algorithm 2 with $R = 1$ and $K = 2$.

This algorithm can be parallelized since we can compute $\tilde{\mathcal{T}}(x^{(k,1)}, ..., x^{(k,R)})_j$ for each $j$ parallelly. It can be shown that it is sufficient to choose $K$ to depend logarithmically on $\kappa$ and $\epsilon$. Similar to Algorithm 1, we can show that Algorithm 2 has the guarantee that the bias of the discretization is of order $\tilde{O}\left(\frac{h^4}{R}\sqrt{\frac{d}{L}}\right)$ and the standard deviation is of order $\tilde{O}\left(\frac{h^2}{R}\sqrt{\frac{d}{L}}\right)$ (Appendix F). Then,

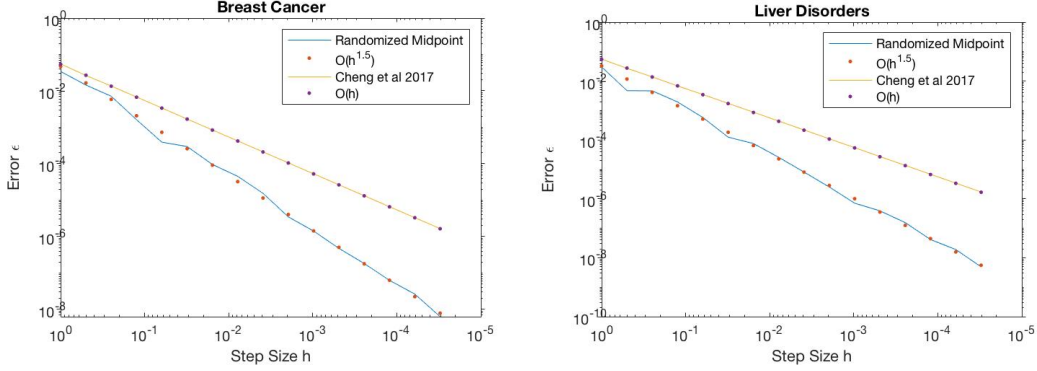

Figure 1: Error of random walks with different choice of step size.

summing from $\tilde{O}\left(\frac{\kappa}{h}\right)$ iterations, the total bias would be $\tilde{O}\left(\frac{h^4}{R}\sqrt{\frac{d}{L}}\cdot\frac{\kappa}{h}\right) = \tilde{O}\left(\frac{h^3}{R}\sqrt{\frac{d\kappa}{m}}\right)$, and

the total standard deviation would be $\tilde{O}\left(\frac{h^2}{R}\sqrt{\frac{d}{L}}\cdot\sqrt{\frac{\kappa}{h}}\right) = \tilde{O}\left(\frac{h^{1.5}}{R}\sqrt{\frac{d}{m}}\right)$. By choosing $R =$

$\tilde{\Theta}\left(\frac{\sqrt{\kappa}}{\epsilon}\right)$, it is enough to choose $h$ to be a constant to achieve less than $\epsilon\sqrt{\frac{d}{m}}$ error, which shows that
the algorithm needs only $O\left(\frac{\kappa}{h}\log\frac{1}{\epsilon}\right) = O(\kappa\log\frac{1}{\epsilon})$ parallel steps. Appendix F gives a partial proof
of the guarantee of Algorithm 2. The other part of the proof is similar to that in Algorithm 1, so we
omit it here.

**Theorem 4.** *Let $f$ be a function such that $0 \prec m \cdot I_d \preceq \nabla^2 f(x) \preceq L \cdot I_d$ for all $x \in \mathbb{R}^d$. Let $Y$
be a random point drawn from the density proportional to $e^{-f}$. Algorithm 2 can generate a random
point $X$ such that $W_2(X,Y) \leq \epsilon\sqrt{\frac{d}{m}}$ in $O(\kappa\log\frac{1}{\epsilon})$ parallel steps. Furthermore, each iteration of
Algorithm 2 involves computing $\tilde{\Theta}\left(\frac{\sqrt{\kappa}}{\epsilon}\right)$ of $\nabla fs$.*

## 5  Numerical Experiments

In this section, we compare the algorithm from our paper, randomized midpoint method, with the one
from [10]. We test the algorithms on the liver-disorders dataset and the breast-cancer dataset from
UCL machine learning [17]. In both datasets, we observe a set of independent samples $\{x_i, y_i\}_{i=1}^m$,
where $y_i$ is the label, $x_i$ is the feature and $m$ is the number of samples. We sample from the target
distribution $p^*(\theta) \propto \exp\left(-f(\theta)\right)$, where

$$f(\theta) = \frac{\lambda}{2}\|\theta\|^2 + \frac{1}{m}\sum_{i=1}^m \log\left(\exp\left(-y_i x_i^T\theta\right) + 1\right),$$

for regularization parameters $\lambda$. We set $\lambda$ to be $10^{-2}$ in our experiments. Figure 1 shows the error
of randomized midpoint method and the algorithm from [10] with different step size $h$. The error
is measured by the $\ell_2$ distance to the true solution of (3) at time $N = 5000$, a time much greater
than the mixing time of (3) for both datasets. Our results show that the $\epsilon$ dependence analysis of our
algorithm and that of [10] are both tight. However, we note that the logistic function is infinitely
differentiable, so there are methods of higher orders for this objective such as the standard midpoint
method and Runge–Kutta methods.

## Footnotes

[1]Previous papers addressing this problem defined $\epsilon$ as $W_2(x, e^{-f}) \leq \epsilon$. This definition is not scale invariant, i.e., the number of steps changes when we scale $f$. In comparison, our definition yields results that are invariant under: (1) the scaling of $f$, namely, replacing $f(x)$ by $\alpha f(x)$ for $\alpha > 0$, and (2) the tensor power of $f$, namely, replacing $f(x)$ by $g(x) \stackrel{\text{def}}{=} \sum_i f(x_i)$. Our new definition of $\epsilon$ also clarifies definitions in previous research. Under the prior definition of $\epsilon$, the algorithms [19, 10, 7] take $\tilde{O}(\kappa^2(\sqrt{\frac{d}{m}}/\epsilon)^2)$, $\tilde{O}(\kappa^2\sqrt{\frac{d}{m}}/\epsilon)$, and $\tilde{O}(\kappa^{1.5}\sqrt{\frac{d}{m}}/\epsilon)$ steps, respectively. Our new definition shows that these different dependences on $d$ and $m$ all relate to their dependence on $\epsilon$.

[2]The corresponding optimization problem takes at least $\min(\sqrt{\kappa}, d)$ steps [3]. If we represent each point the optimization algorithm visited by a vertex and each step the algorithm takes by an edge, then the existing lower bound in fact shows that this graph has a diameter of at least $\min(\sqrt{\kappa}, d)$. Since a random walk on a graph of diameter $D$ takes $D^2$ to mix, a random walk on the graph takes at least $\min(\sqrt{\kappa}, d)^2$ steps.

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
