[Supplementary Material · RMM_Appendix.pdf]

# A  Brownian Motion Simulation

In this section, we introduce how $W_1$, $W_2$ and $W_3$ can be sampled. Let $\{B_t\}_{t \in [0,h]}$ be the standard $d$-dimensional Brownian motion on $t \in [0, h]$. In Algorithm 1, $W_1 = \int_0^{\alpha h} \left(1 - e^{-2(\alpha h - s)}\right) \mathrm{d}B_s$, $W_2 = \int_0^h \left(1 - e^{-2(h-s)}\right) \mathrm{d}B_s$ and $W_3 = \int_0^h e^{-2(h-s)} \mathrm{d}B_s$. We define $G_1 = \int_0^{\alpha h} e^{2s} \mathrm{d}B_s$, $G_2 = \int_{\alpha h}^h e^{2s} \mathrm{d}B_s$, $H_1 = \int_0^{\alpha h} \mathrm{d}B_s$ and $H_2 = \int_{\alpha h}^h \mathrm{d}B_s$. Then, $W_1 = H_1 - e^{-2\alpha h} G_1$, $W_2 = (H_1 + H_2) - e^{-2h}(G_1 + G_2)$ and $W_3 = e^{-2h}(G_1 + G_2)$. It is sufficient to sample $H_1$, $H_2$, $G_1$ and $G_2$. We can show that $(G_1, H_1)$ is independent of $(G_2, H_2)$, and $(G_1, H_1)$ and $(G_2, H_2)$ both follow a $2d$-dimensional Gaussian distribution, which can be easily sampled.

**Lemma 5.** *Define $G_1 = \int_0^{\alpha h} e^{2s} \mathrm{d}B_s$, $G_2 = \int_{\alpha h}^h e^{2s} \mathrm{d}B_s$, $H_1 = \int_0^{\alpha h} \mathrm{d}B_s$ and $H_2 = \int_{\alpha h}^h \mathrm{d}B_s$. Then, $(G_1, H_1)$ is independent of $(G_2, H_2)$. Moreover, $(G_1, H_1)$ and $(G_2, H_2)$ both follow a 2d-dimensional Gaussian distribution with mean zero. Conditional on the choice of $\alpha$, their covariance is given by*

$$\mathbb{E}\left[(G_1 - \mathbb{E}G_1)(H_1 - \mathbb{E}H_1)^T\right] = \frac{1}{2}\left(e^{2\alpha h} - 1\right) \cdot I_d,$$

$$\mathbb{E}\left[(G_1 - \mathbb{E}G_1)(G_1 - \mathbb{E}G_1)^T\right] = \frac{1}{4}\left(e^{4\alpha h} - 1\right) \cdot I_d,$$

$$\mathbb{E}\left[(H_1 - \mathbb{E}H_1)(H_1 - \mathbb{E}H_1)^T\right] = \alpha h \cdot I_d,$$

$$\mathbb{E}\left[(G_2 - \mathbb{E}G_2)(H_2 - \mathbb{E}H_2)^T\right] = \frac{1}{2}\left(e^{2h} - e^{2\alpha h}\right) \cdot I_d,$$

$$\mathbb{E}\left[(G_2 - \mathbb{E}G_2)(G_2 - \mathbb{E}G_2)^T\right] = \frac{1}{4}\left(e^{4h} - e^{4\alpha h}\right) \cdot I_d,$$

$$\mathbb{E}\left[(H_2 - \mathbb{E}H_2)(H_2 - \mathbb{E}H_2)^T\right] = (h - \alpha h) \cdot I_d.$$

*Proof.* By the definition of the standard Brownian motion, $(G_1, H_1)$ is independent of $(G_2, H_2)$ and $(G_1, H_1)$ and $(G_2, H_2)$ both have mean zero. Moreover,

$$\mathbb{E}\left[(G_1 - \mathbb{E}G_1)(H_1 - \mathbb{E}H_1)^T\right] = \mathbb{E}\left[\left(\int_0^{\alpha h} e^{2s} \mathrm{d}B_s\right)\left(\int_0^{\alpha h} \mathrm{d}B_s\right)^T\right] = \int_0^{\alpha h} e^{2s} \mathrm{d}s \cdot I_d$$

$$= \frac{1}{2}\left(e^{2\alpha h} - 1\right) \cdot I_d,$$

$$\mathbb{E}\left[(G_1 - \mathbb{E}G_1)(G_1 - \mathbb{E}G_1)^T\right] = \mathbb{E}\left[\left(\int_0^{\alpha h} e^{2s} \mathrm{d}B_s\right)\left(\int_0^{\alpha h} e^{2s} \mathrm{d}B_s\right)^T\right] = \int_0^{\alpha h} e^{4s} \mathrm{d}s \cdot I_d$$

$$= \frac{1}{4}\left(e^{4\alpha h} - 1\right) \cdot I_d,$$

and

$$\mathbb{E}\left[(H_1 - \mathbb{E}H_1)(H_1 - \mathbb{E}H_1)^T\right] = \alpha h \cdot I_d.$$

Similarly,

$$\mathbb{E}\left[(G_2 - \mathbb{E}G_2)(H_2 - \mathbb{E}H_2)^T\right] = \mathbb{E}\left[\left(\int_{\alpha h}^h e^{2s} \mathrm{d}B_s\right)\left(\int_{\alpha h}^h \mathrm{d}B_s\right)^T\right] = \int_{\alpha h}^h e^{2s} \mathrm{d}s \cdot I_d$$

$$= \frac{1}{2}\left(e^{2h} - e^{2\alpha h}\right) \cdot I_d,$$

$$\mathbb{E}\left[(G_1 - \mathbb{E}G_1)(G_1 - \mathbb{E}G_1)^T\right] = \mathbb{E}\left[\left(\int_{\alpha h}^h e^{2s} \mathrm{d}B_s\right)\left(\int_{\alpha h}^h e^{2s} \mathrm{d}B_s\right)^T\right] = \int_{\alpha h}^h e^{4s} \mathrm{d}s \cdot I_d$$

$$= \frac{1}{4} \left( e^{4h} - e^{4\alpha h} \right) \cdot I_d,$$

and

$$\mathbb{E} \left[ \left( H_2 - \mathbb{E} H_2 \right) \left( H_2 - \mathbb{E} H_2 \right)^T \right] = (h - \alpha h) \cdot I_d.$$

$\square$

## B  Properties of the ULD and the Brownian motion

Here, we prove some properties of the ULD and the Brownian motion. These properties are used in Appendices C, D, E and F to prove the guarantee of our algorithm.

### B.1  Properties of the ULD

**Lemma 6.** *Let* $\{x(t)\}_{t \in [0,h]}$ *and* $\{v(t)\}_{t \in [0,h]}$ *be the solution to the underdamped Langevin diffusion* (3) *on* $t \in [0, h]$. *Assume that* $h \leq \frac{1}{20}$ *and* $u = \frac{1}{L}$. *We have the following bounds.*

$$\mathbb{E} \sup_{t \in [0,h]} \|v(t)\|^2 \leq O \left( \|v(0)\|^2 + u^2 h^2 \|\nabla f(x(0))\|^2 + udh \right),$$

$$\mathbb{E} \sup_{t \in [0,h]} \|\nabla f(x(t))\|^2 \leq O \left( \|\nabla f(x(0))\|^2 + L^2 h^2 \|v(0)\|^2 + Ldh^3 \right),$$

$$\mathbb{E} \sup_{t \in [0,h]} \|x(0) - x(t)\|^2 \leq O \left( h^2 \|v(0)\|^2 + u^2 h^4 \|\nabla f(x(0))\|^2 + udh^3 \right),$$

*and*

$$-\mathbb{E} \inf_{t \in [0,h]} \|v(t)\|^2 \leq -\frac{1}{3} \|v(0)\|^2 + O \left( u^2 h^2 \|\nabla f(x(0))\|^2 + udh \right),$$

$$-\mathbb{E} \inf_{t \in [0,h]} \|\nabla f(x(t))\|^2 \leq -\frac{1}{3} \|\nabla f(x(0))\|^2 + O \left( h^2 L^2 \|v(0)\|^2 + Ldh^3 \right).$$

*Proof.* We first show the first three bounds. We can write $\mathbb{E} \sup_{t \in [0,h]} \|\nabla f(x(t))\|^2$ as

$$\mathbb{E} \sup_{t \in [0,h]} \|\nabla f(x(t))\|^2$$
$$\leq 2 \|\nabla f(x(0))\|^2 + 2 \mathbb{E} \sup_{t \in [0,h]} \|\nabla f(x(0)) - \nabla f(x(t))\|^2$$
$$\leq 2 \|\nabla f(x(0))\|^2 + 2L^2 \mathbb{E} \sup_{t \in [0,h]} \|x(0) - x(t)\|^2, \tag{6}$$

where the first step follows by Young's inequality and the second step follows by $\nabla f$ is $L$-Lipschitz. To bound $\mathbb{E} \sup_{t \in [0,h]} \|x(0) - x(t)\|^2$,

$$\mathbb{E} \sup_{t \in [0,h]} \|x(0) - x(t)\|^2 = \mathbb{E} \sup_{t \in [0,h]} \left\| \int_0^t v(s) \, \mathrm{d}s \right\|^2$$
$$\leq \mathbb{E} \sup_{t \in [0,h]} t \int_0^t \|v(s)\|^2 \, \mathrm{d}s$$
$$\leq h^2 \mathbb{E} \sup_{t \in [0,h]} \|v(t)\|^2, \tag{7}$$

where the first step follows by the definition of $x$ and the second follows by the Cauchy-Schwarz inequality. To bound $\mathbb{E} \sup_{t \in [0,h]} \|v(t)\|^2$,

$$\mathbb{E} \sup_{t \in [0,h]} \|v(t)\|^2 = \mathbb{E} \sup_{t \in [0,h]} \left\| v(0) e^{-2t} - u \int_0^t e^{-2(t-s)} \nabla f(x(s)) \, \mathrm{d}s + 2\sqrt{u} \int_0^t e^{-2(t-s)} \, \mathrm{d}B_s \right\|^2$$

$$\leq \quad 3\left\|v(0)\right\|^2 + 3u^2h^2\mathbb{E}\sup_{t\in[0,h]}\left\|\nabla f(x(t))\right\|^2 + 12u\mathbb{E}\sup_{t\in[0,h]}\left\|\int_0^t e^{-2(t-s)}\,\mathrm{d}B_s\right\|^2$$

$$\leq \quad 3\left\|v(0)\right\|^2 + 3u^2h^2\mathbb{E}\sup_{t\in[0,h]}\left\|\nabla f(x(t))\right\|^2 + 60udh, \tag{8}$$

where the first step follows by the definition of ULD, the second step follows by the inequality $(a+b+c)^2 \leq 3a^2 + 3b^2 + 3c^2$ and the third step follows by Lemma 8. Then, combining (6), (7) and (8), we have

$$\mathbb{E}\sup_{t\in[0,h]}\left\|\nabla f(x(t))\right\|^2 \leq 2\left\|\nabla f(x(0))\right\|^2 + 2L^2\mathbb{E}\sup_{t\in[0,h]}\left\|x(0)-x(t)\right\|^2$$

$$\leq 2\left\|\nabla f(x(0))\right\|^2 + 2L^2h^2\mathbb{E}\sup_{t\in[0,h]}\left\|v(t)\right\|^2$$

$$\leq 2\left\|\nabla f(x(0))\right\|^2 + 6h^4\mathbb{E}\sup_{t\in[0,h]}\left\|\nabla f(x(t))\right\|^2 + 6L^2h^2\left\|v(0)\right\|^2 + 120Ldh^3.$$

Since $6h^4 \leq \frac{1}{4}$,

$$\mathbb{E}\sup_{t\in[0,h]}\left\|\nabla f(x(t))\right\|^2 \quad \leq \quad 3\left\|\nabla f(x(0))\right\|^2 + 8L^2h^2\left\|v(0)\right\|^2 + 160Ldh^3$$

$$\leq \quad O\left(\left\|\nabla f(x(0))\right\|^2 + L^2h^2\left\|v(0)\right\|^2 + Ldh^3\right). \tag{9}$$

By (8) and (9),

$$\mathbb{E}\sup_{t\in[0,h]}\left\|v(t)\right\|^2 \quad \leq \quad 3\left\|v(0)\right\|^2 + 3u^2h^2\mathbb{E}\sup_{t\in[0,h]}\left\|\nabla f(x(t))\right\|^2 + 60udh$$

$$\leq \quad 3\left\|v(0)\right\|^2 + 3u^2h^2\cdot O\left(\left\|\nabla f(x(0))\right\|^2 + L^2h^2\left\|v(0)\right\|^2 + Ldh^3\right) + 60udh$$

$$\leq \quad O\left(\left\|v(0)\right\|^2 + u^2h^2\left\|\nabla f(x(0))\right\|^2 + udh\right).$$

where the last step follows by $h$ is small.

By (7) and (9),

$$\mathbb{E}\sup_{t\in[0,h]}\left\|x(0)-x(t)\right\|^2 \quad \leq \quad h^2\mathbb{E}\sup_{t\in[0,h]}\left\|v(t)\right\|^2$$

$$\leq \quad O\left(h^2\left\|v(0)\right\|^2 + u^2h^4\left\|\nabla f(x(0))\right\|^2 + udh^3\right). \tag{10}$$

To prove the fourth claim,

$$\inf_{t\in[0,h]}\left\|v(t)\right\|^2$$

$$= \quad \inf_{t\in[0,h]}\left\|v(0)e^{-2t} - u\int_0^t e^{-2(t-s)}\nabla f(x(s))\,\mathrm{d}s + 2\sqrt{u}\int_0^t e^{-2(t-s)}\,\mathrm{d}B_s\right\|^2$$

$$\geq \quad \inf_{t\in[0,h]}\left[e^{-4t}\left\|v(0)\right\|^2 - 2e^{-2t}v(0)^T\left(u\int_0^t e^{-2(t-s)}\nabla f(x(s))\,\mathrm{d}s\right)\right.$$

$$\left. + 2e^{-2t}v(0)^T\left(2\sqrt{u}\int_0^t e^{-2(t-s)}\,\mathrm{d}B_s\right)\right]$$

$$\geq \quad \inf_{t\in[0,h]}\left[e^{-4t}\left\|v(0)\right\|^2 - \frac{1}{2}e^{-4t}\left\|v(0)\right\|^2 - 4\left\|u\int_0^t e^{-2(t-s)}\nabla f(x(s))\,\mathrm{d}s\right\|^2\right.$$

$$\left. - 4\left\|2\sqrt{u}\int_0^t e^{-2(t-s)}\,\mathrm{d}B_s\right\|^2\right]$$

$$\geq \quad \inf_{t\in[0,h]}\left[\frac{1}{2}(1-4h)\left\|v(0)\right\|^2 - 4u^2h^2\sup_{s\in[0,t]}\left\|\nabla f(x(s))\right\|^2 - 16u\left\|\int_0^t e^{-2(t-s)}\,\mathrm{d}B_s\right\|^2\right]$$

$$\geq \quad \frac{1}{2}(1-4h)\|v(0)\|^2 - 4u^2h^2 \sup_{t\in[0,h]} \|\nabla f(x(t))\|^2 - 16u \sup_{t\in[0,h]} \left\|\int_0^t e^{-2(t-s)}\,\mathrm{d}B_s\right\|^2,$$

where the first step follows by the definition of $v$, the second step follows by the inequality $(a + b + c)^2 \geq a^2 + 2a(b+c)$, the third step follows by the inequality $2ab \leq a^2 + b^2$, the fourth step follows by $e^{-4t} \geq 1 - 4t$, and the last step follows by $h$ is small.

Then, by (9) and Lemma 8,

$$-\mathbb{E}\inf_{t\in[0,h]}\|v(t)\|^2 \quad \leq \quad -\frac{1}{3}\|v(0)\|^2 + O\left(u^2h^2\|\nabla f(x(0))\|^2 + udh\right).$$

To show the lower bound on $\mathbb{E}\inf_{t\in[0,h]}\|\nabla f(x(t))\|^2$, notice that

$$\mathbb{E}\inf_{t\in[0,h]}\|\nabla f(x(t))\|^2 \quad \geq \quad \frac{1}{2}\|\nabla f(x(0))\|^2 - \mathbb{E}\sup_{t\in[0,h]}\|\nabla f(x(t)) - \nabla f(x(0))\|^2$$

$$\geq \quad \frac{1}{2}\|\nabla f(x(0))\|^2 - L^2\mathbb{E}\sup_{t\in[0,h]}\|x(t) - x(0)\|^2.$$

Then, by (10) and $h \leq \frac{1}{20}$,

$$-\mathbb{E}\inf_{t\in[0,h]}\|\nabla f(x(t))\| \quad \leq \quad -\frac{1}{3}\|\nabla f(x(0))\|^2 + O\left(h^2L^2\|v(0)\|^2 + Ldh^3\right).$$

$\square$

## B.2 Properties of the Brownian Motion

**Lemma 7** (Doob's maximal inequality [16]). *Suppose $\{X(t) : t \geq 0\}$ is a continuous martingale. Then, for any $t \geq 0$,*

$$\mathbb{E}\left[\sup_{0\leq s\leq t}|X(s)|^2\right] \quad \leq \quad 4\mathbb{E}\left[|X(t)|^2\right].$$

Using the Doob's maximal inequality, we can show the following lemma.

**Lemma 8.** *For $d$-dimensional Brownian motion $B_t$ on $t \in [0, h]$, assuming $h \leq \frac{1}{10}$,*

$$\mathbb{E}\left[\sup_{0\leq t\leq h}\|B(t)\|^2\right] \leq 4dh, \text{ and } \mathbb{E}\left[\sup_{0\leq t\leq h}\left\|\int_0^t e^{-2(t-s)}dB_s\right\|^2\right] \leq 5dh.$$

*Proof.* To show the first inequality,

$$\mathbb{E}\left[\sup_{0\leq t\leq h}\|B(t)\|^2\right] \quad \leq \quad \sum_{i=1}^d \mathbb{E}\left[\sup_{0\leq t\leq h}|B_i(t)|^2\right]$$

$$\leq \quad 4d\mathbb{E}\left[|B_i(h)|^2\right]$$

$$= \quad 4dh,$$

where the second step follows by Lemma 7. To show the second inequality,

$$\mathbb{E}\left[\sup_{0\leq t\leq h}\left\|\int_0^t e^{-2(t-s)}dB_s\right\|^2\right] \quad \leq \quad \mathbb{E}\left[\sup_{0\leq t\leq h}e^{-4t}\left\|\int_0^t e^{2s}dB_s\right\|^2\right]$$

$$\leq \quad \mathbb{E}\left[\sup_{0\leq t\leq h}\left\|\int_0^t e^{2s}dB_s\right\|^2\right]$$

$$\leq \quad \sum_{i=1}^d \mathbb{E}\left[\sup_{0\leq t\leq h}\left|\int_0^t e^{2s}dB_{s,i}\right|^2\right]$$

$$\leq \quad 4\sum_{i=1}^{d} \mathbb{E}\left[\left|\int_0^h e^{2s}dB_{s,i}\right|^2\right]$$

$$= \quad 4\sum_{i=1}^{d} \int_0^h e^{4s}ds$$

$$\leq \quad 5dh,$$

where the second step follows by $e^{-4t} \leq 1$, the fourth step follows by Lemma 7 and the last inequality follows by $\int_0^h e^{4s}ds \leq \frac{5}{4}h$ for $h \leq \frac{1}{10}$. $\qquad\qquad\square$

## C   Discretization Error of Algorithm 1

In this section, we bound the discretization error of Algorithm 1 in each iteration. In order to prove Lemma 2, we first prove Lemma 9, stated next.

**Lemma 9.** *Let $\alpha$ be the random number chosen in iteration $n$. Let $x_{n+\frac{1}{2}}$ be the intermediate value computed in iteration $n$ of Algorithm 1. Let $\{x_n^*(t)\}_{t\in[0,h]}$ be the ideal underdamped Langevin diffusion starting from $x_n^*(0) = x_n$ coupled through a shared Brownian motion with $x_{n+\frac{1}{2}}$. Assume that $h \leq \frac{1}{20}$. Then,*

$$\mathbb{E}\left\|\nabla f(x_{n+\frac{1}{2}}) - \nabla f(x_n^*(\alpha h))\right\|^2 \quad \leq \quad O\left(h^6 L^2 \|v_n\|^2 + h^8 \|\nabla f(x_n)\|^2 + Ldh^7\right).$$

*Proof.* We have the bound

$$\mathbb{E}\left\|\nabla f(x_{n+\frac{1}{2}}) - \nabla f(x_n^*(\alpha h))\right\|^2$$

$$\leq \quad L^2 \mathbb{E}\left\|x_{n+\frac{1}{2}} - x_n^*(\alpha h)\right\|^2$$

$$= \quad L^2 \mathbb{E}\left\|\frac{1}{2}u\int_0^{\alpha h}\left(1 - e^{-2(\alpha h - s)}\right)\left(\nabla f(x_n^*(0)) - \nabla f(x_n^*(s))\right)ds\right\|^2$$

$$\leq \quad \frac{1}{4}\mathbb{E}\left[\int_0^{\alpha h}\left(1 - e^{-2(\alpha h - s)}\right)^2 ds \cdot \alpha h \cdot \left(\sup_{t\in[0,h]}\|\nabla f(x_n^*(0)) - \nabla f(x_n^*(t))\|^2\right)\right]$$

$$\leq \quad h^4 \mathbb{E}\sup_{t\in[0,h]}\|\nabla f(x_n^*(0)) - \nabla f(x_n^*(t))\|^2$$

$$\leq \quad L^2 h^4 \mathbb{E}\sup_{t\in[0,h]}\|x_n^*(0) - x_n^*(t)\|^2$$

$$\leq \quad O\left(h^6 L^2 \|v_n\|^2 + h^8 \|\nabla f(x_n)\|^2 + Ldh^7\right),$$

where the first and the fifth step follows by $\nabla f$ is $L$-Lipschitz, the third step follows by Cauchy-Schwarz inequality, the fourth step follows by $1 - e^{-2(\alpha h - t)} \leq 2h$ and the last step follows by Lemma 6. $\qquad\square$

Now, we are ready to prove Lemma 2.

*Proof.* To show the first claim,

$$\|\mathbb{E}_\alpha x_{n+1} - x_n^*(h)\|^2$$

$$= \quad \left\|\mathbb{E}_\alpha \frac{1}{2}uh\left(1 - e^{-2(h-\alpha h)}\right)\nabla f(x_{n+\frac{1}{2}}) - \frac{1}{2}u\int_0^h\left(1 - e^{-2(h-s)}\right)\nabla f(x_n^*(s))ds\right\|^2$$

$$\leq \quad \frac{1}{2}\mathbb{E}_\alpha\left\|uh\left(1 - e^{-2(h-\alpha h)}\right)\nabla f(x_{n+\frac{1}{2}}) - uh\left(1 - e^{-2(h-\alpha h)}\right)\nabla f(x_n^*(\alpha h))\right\|^2$$

$$+\frac{1}{2}\left\|\mathbb{E}_\alpha uh\left(1-e^{-2(h-\alpha h)}\right)\nabla f(x_n^*(\alpha h))-u\int_0^h\left(1-e^{-2(h-s)}\right)\nabla f(x_n^*(s))\,\mathrm{d}s\right\|^2$$

$$\leq\quad\frac{1}{2}u^2h^2\mathbb{E}_\alpha\left[\left(1-e^{-2(h-\alpha h)}\right)^2\left\|\nabla f(x_{n+\frac{1}{2}})-\nabla f(x_n^*(\alpha h))\right\|^2\right]+0$$

$$\leq\quad 2u^2h^4\mathbb{E}_\alpha\left\|\nabla f(x_{n+\frac{1}{2}})-\nabla f(x_n^*(\alpha h))\right\|^2,$$

where the first step follows by the definition of $x_{n+1}$, the second step follows by Young's inequality, the third step follows by

$$\mathbb{E}_\alpha h\left(1-e^{-2(h-\alpha h)}\right)\nabla f(x_n^*(\alpha h))=\int_0^h\left(1-e^{-2(h-s)}\right)\nabla f(x_n^*(s))\,\mathrm{d}s,$$

and the fourth step follows by $1-e^{-2(h-\alpha h)}\leq 2h$ . By Lemma 9,

$$\mathbb{E}\left\|\mathbb{E}_\alpha x_{n+1}-x_n^*(h)\right\|^2\quad\leq\quad O\left(h^{10}\left\|v_n\right\|^2+u^2h^{12}\left\|\nabla f(x_n)\right\|^2+udh^{11}\right).$$

To show the second claim,

$$\mathbb{E}\left\|x_{n+1}-x_n^*(h)\right\|^2$$

$$\leq\quad\frac{3}{4}\mathbb{E}\left\|uh\left(1-e^{-2(h-\alpha h)}\right)\nabla f(x_{n+\frac{1}{2}})-uh\left(1-e^{-2(h-\alpha h)}\right)\nabla f(x_n^*(\alpha h))\right\|^2$$

$$+\frac{3}{4}\mathbb{E}\left\|uh\left(1-e^{-2(h-\alpha h)}\right)\nabla f(x_n^*(\alpha h))-u\int_0^h\left(1-e^{-2(h-\alpha h)}\right)\nabla f(x_n^*(s))\,\mathrm{d}s\right\|^2$$

$$+\frac{3}{4}\mathbb{E}\left\|u\int_0^h\left(1-e^{-2(h-\alpha h)}\right)\nabla f(x_n^*(s))\,\mathrm{d}s-u\int_0^h\left(1-e^{-2(h-s)}\right)\nabla f(x_n^*(s))\,\mathrm{d}s\right\|^2,$$

which follows by definition and Young's inequality. To bound the second term,

$$\left\|uh\left(1-e^{-2(h-\alpha h)}\right)\nabla f(x_n^*(\alpha h))-u\int_0^h\left(1-e^{-2(h-\alpha h)}\right)\nabla f(x_n^*(s))\,\mathrm{d}s\right\|^2$$

$$=\quad\left\|u\int_0^h\left(1-e^{-2(h-\alpha h)}\right)\left(\nabla f(x_n^*(\alpha h))-\nabla f(x_n^*(s))\right)\,\mathrm{d}s\right\|^2$$

$$\leq\quad u^2\int_0^h\left(1-e^{-2(h-\alpha h)}\right)^2\,\mathrm{d}s\cdot\sup_{t\in[0,h]}\left\|\nabla f(x_n^*(\alpha h))-\nabla f(x_n^*(t))\right\|^2\cdot h$$

$$\leq\quad 4u^2h^4\sup_{t\in[0,h]}\left\|\nabla f(x_n^*(\alpha h))-\nabla f(x_n^*(t))\right\|^2$$

$$\leq\quad 16h^4\sup_{t\in[0,h]}\left\|x_n^*(0)-x_n^*(t)\right\|^2\tag{11}$$

where the second step follows by the Cauchy-Schwarz inequality. The third term satisfies

$$\left\|u\int_0^h\left(1-e^{-2(h-\alpha h)}\right)\nabla f(x_n^*(s))\,\mathrm{d}s-u\int_0^h\left(1-e^{-2(h-s)}\right)\nabla f(x_n^*(s))\,\mathrm{d}s\right\|^2$$

$$=\quad u^2\left\|\int_0^h\left(e^{-2(h-s)}-e^{-2(h-\alpha h)}\right)\nabla f(x_n^*(s))\,\mathrm{d}s\right\|^2$$

$$\leq\quad 4u^2h^4\sup_{t\in[0,h]}\left\|\nabla f(x_n^*(t))\right\|^2,\tag{12}$$

where the second step follows by the Cauchy Schwarz inequality and $\left|e^{-2(h-s)}-e^{-2(h-\alpha h)}\right|\leq 2h$. Thus,

$$\mathbb{E}\left\|x_{n+1}-x_n^*(h)\right\|^2$$

$$
\begin{aligned}
\leq\quad & 3u^2 h^4 \mathbb{E}\left\|\nabla f(x_{n+\frac{1}{2}}) - \nabla f(x_n^*(\alpha h))\right\|^2 + 12 h^4 \mathbb{E}\sup_{t\in[0,h]} \|x_n^*(0) - x_n^*(t)\|^2 \\
& + 3u^2 h^4 \mathbb{E}\sup_{t\in[0,h]} \|\nabla f(x_n^*(t))\|^2 \\
\leq\quad & 3h^4 \cdot O\left(h^6 \|v_n\|^2 + h^8 u^2 \|\nabla f(x_n)\|^2 + u d h^7\right) \\
& + 12 h^4 \cdot O\left(h^2 \|v_n\|^2 + u^2 h^4 \|\nabla f(x_n)\|^2 + u d h^3\right) \\
& + 3u^2 h^4 \cdot O\left(\|\nabla f(x_n)\|^2 + L^2 h^2 \|v_n\|^2 + M d h^3\right) \\
\leq\quad & O\left(h^6 \|v_n\|^2 + u^2 h^4 \|\nabla f(x_n)\|^2 + u d h^7\right).
\end{aligned}
$$

where the first step follows by (11) and (12), the second step follows by Lemma 6 and Lemma 9, and the last inequality follows by $h \leq 1$.

To show the third claim,

$$
\begin{aligned}
\mathbb{E}\left\|\mathbb{E}_\alpha v_{n+1} - v_n^*(h)\right\|^2 \ =\ & \mathbb{E}\left\|\mathbb{E}_\alpha u h e^{-2(h-\alpha h)}\nabla f(x_{n+\frac{1}{2}}) - u\int_0^h e^{-2(h-s)}\nabla f(x_n^*(s))\,\mathrm{d}s\right\|^2 \\
\leq\ & 2\mathbb{E}\left\|u h e^{-2(h-\alpha h)}\nabla f(x_{n+\frac{1}{2}}) - u h e^{-2(h-\alpha h)}\nabla f(x_n^*(\alpha h))\right\|^2 \\
& + 2\mathbb{E}\left\|\mathbb{E}_\alpha u h e^{-2(h-\alpha h)}\nabla f(x_n^*(\alpha h)) - u\int_0^h e^{-2(h-s)}\nabla f(x_n^*(s))\,\mathrm{d}s\right\|^2 \\
\leq\ & 2u^2 h^2 \mathbb{E}\left\|\nabla f(x_{n+\frac{1}{2}}) - \nabla f(x_n^*(\alpha h))\right\|^2 + 0 \\
\leq\ & O\left(h^8 \|v_n\|^2 + u^2 h^{10} \|\nabla f(x_n)\|^2 + u d h^9\right),
\end{aligned}
$$

where the first step follows by Young's inequality, the second step follows by

$$
\mathbb{E}_\alpha u h e^{-2(h-\alpha h)}\nabla f(x_n^*(\alpha h)) \ =\ u\int_0^h e^{-2(h-t)}\nabla f(x_n^*(t))\,\mathrm{d}t,
$$

and $e^{-2(h-\alpha h)} \leq 1$, and the third step follows by Lemma 9.

To show the last claim,

$$
\begin{aligned}
& \mathbb{E}\left\|v_{n+1} - v_n^*(h)\right\|^2 \\
=\ & \mathbb{E}\left\|u h e^{-2(h-\alpha h)}\nabla f(x_{n+\frac{1}{2}}) - u\int_0^h e^{-2(h-s)}\nabla f(x^*(s))\,\mathrm{d}s\right\|^2 \\
\leq\ & 3\mathbb{E}\left\|u h e^{-2(h-\alpha h)}\nabla f(x_{n+\frac{1}{2}}) - u h e^{-2(h-\alpha h)}\nabla f(x^*(\alpha h))\right\|^2 \\
& + 3\mathbb{E}\left\|u\int_0^h e^{-2(h-\alpha h)}\nabla f(x_n^*(\alpha h))\,\mathrm{d}t - u\int_0^h e^{-2(h-\alpha h)}\nabla f(x_n^*(s))\,\mathrm{d}s\right\|^2 \\
& + 3\mathbb{E}\left\|u\int_0^h e^{-2(h-\alpha h)}\nabla f(x_n^*(s))\,\mathrm{d}s - u\int_0^h e^{-2(h-s)}\nabla f(x_n^*(s))\,\mathrm{d}s\right\|^2 \\
\leq\ & 3u^2 h^2 \mathbb{E}\left\|\nabla f(x_{n+\frac{1}{2}}) - \nabla f(x_n^*(\alpha h))\right\|^2 + 3h^2 \mathbb{E}\sup_{t\in[0,h]} \|x_n^*(\alpha h) - x_n^*(t)\|^2 \\
& + 12 u^2 h^4 \mathbb{E}\sup_{t\in[0,h]} \|\nabla f(x_n^*(t))\|^2 \\
\leq\ & 3u^2 h^2 \cdot O\left(h^6 L^2 \|v_n\|^2 + h^8 \|\nabla f(x_n)\|^2 + L d h^7\right) \\
& + 3h^2 \cdot O\left(h^2 \|v_n\|^2 + u^2 h^4 \|\nabla f(x_n)\|^2 + u d h^3\right)
\end{aligned}
$$

$$+12u^2h^4 \cdot O\left(\|\nabla f(x_n)\|^2 + L^2h^2\|v_n\|^2 + Ldh^3\right)$$

$$\leq\quad O\left(h^4\|v_n\|^2 + u^2h^4\|\nabla f(x_n)\|^2 + udh^5\right),$$

where the first step follows by the definition, the second step follows by Young's inequality, the third follows by $e^{-2(h-\alpha h)} - e^{-2(h-s)} \leq 2h$, the fourth step follows by Lemma 9 and Lemma 6 and the last inequality follows by $h \leq 1$. $\qquad\square$

## D  Bounds on $\|\nabla f(x)\|$ and $\|v\|$

In this section, we bound the sum of $\|\nabla f(x_n)\|^2$ and $\|v_n\|^2$ over all iterations $n$, $\sum_{n=0}^{N-1} \mathbb{E}\|\nabla f(x_n)\|^2$ and $\sum_{n=0}^{N-1} \mathbb{E}\|v_n\|^2$. In Appendix E, we use the results in this appendix together with Lemma 2 to prove the guarantee of our algorithm.

**Lemma 10.** *Assume $h \leq \frac{1}{20}$. For each iteration $n$, let $x_n$ be the starting point of iteration $n$ of Algorithm 1. Let $\{v_n(t), x_n(t)\}_{t \in [0,h]}$ be the solution of the exact underdamped Langevin diffusion starting from $(v_n, x_n)$. Let $\mathbb{E}_\alpha$ be the expectation over the random choice of $\alpha$ in iteration $n$. Then, the difference between the value of $f$ on the starting point of iteration $n+1$, $x_{n+1}$, and that of $x_n(h)$ satisfies*

$$\mathbb{E}f(x_{n+1}(0)) - f(x_n(h)) \leq O\left(uh^3\|\nabla f(x_n(0))\|^2 + Lh^5\|v_n(0)\|^2 + dh^6\right).$$

*Proof.* We first consider the expectation over the choice of $\alpha$ in iteration $n$,

$$\mathbb{E}_\alpha f(x_{n+1}(0))$$

$$\leq\quad f(x_n(h)) + \nabla f(x_n(h))^T\left(\mathbb{E}_\alpha x_{n+1}(0) - x_n(h)\right) + \frac{L}{2}\mathbb{E}_\alpha\|x_{n+1}(0) - x_n(h)\|^2$$

$$\leq\quad f(x_n(h)) + \|\nabla f(x_n(h))\|\,\|\mathbb{E}_\alpha x_{n+1}(0) - x_n(h)\| + \frac{L}{2}\mathbb{E}_\alpha\|x_{n+1}(0) - x_n(h)\|^2$$

$$\leq\quad f(x_n(h)) + uh^3\|\nabla f(x_n(h))\|^2 + \frac{L}{h^3}\|\mathbb{E}_\alpha x_{n+1}(0) - x_n(h)\|^2 + \frac{L}{2}\mathbb{E}_\alpha\|x_{n+1}(0) - x_n(h)\|^2,$$

where the first step follows by $\nabla f$ is $L$-Lipschitz, the second step follows by Cauchy-Schwarz inequality and the third step follows by Young's inequality. By Lemma 2 and Lemma 6,

$$\mathbb{E}f(x_{n+1}(0)) \leq \mathbb{E}f(x_n(h)) + uh^3\mathbb{E}\|\nabla f(x_n(h))\|^2 + \frac{L}{h^3}\mathbb{E}\|\mathbb{E}_\alpha x_{n+1}(0) - x_n(h)\|^2$$

$$+\frac{L}{2}\mathbb{E}\|x_{n+1}(0) - x_n(h)\|^2$$

$$\leq \mathbb{E}f(x_n(h)) + \mathbb{E}uh^3 \cdot O\left(\|\nabla f(x_n(0))\|^2 + L^2h^2\|v_n(0)\|^2 + Ldh^3\right)$$

$$+\mathbb{E}\frac{L}{h^3} \cdot O\left(h^{10}\|v_n(0)\|^2 + u^2h^{12}\|\nabla f(x_n(0))\|^2 + udh^{11}\right)$$

$$+\mathbb{E}\frac{L}{2} \cdot O\left(h^6\|v_n(0)\|^2 + h^4u^2\|\nabla f(x_n(0))\|^2 + udh^7\right)$$

$$\leq \mathbb{E}f(x_n(h)) + O\left(uh^3\mathbb{E}\|\nabla f(x_n(0))\|^2 + Lh^5\mathbb{E}\|v_n(0)\|^2 + dh^6\right).$$

where the second step follows by Lemma 2 and Lemma 6, and the last step follows by $h \leq \frac{1}{20}$. $\quad\square$

**Lemma 11.** *Assume $h$ is smaller than some given constant. For each iteration $n = 0, ..., N-1$, let $(v_n, x_n)$ be the starting point of Algorithm 1 in iteration $n$. Then,*

$$\sum_{n=0}^{N-1}\mathbb{E}\|v_n\|^2 \leq O\left(u^2h\sum_{n=0}^{N-1}\mathbb{E}\|\nabla f(x_n)\|^2 + Nud\right).$$

*Proof.* Let $\{v_n(t), x_n(t)\}_{t \in [0,h]}$ be the solution of the exact underdamped Langevin diffusion starting from $(v_n, x_n)$. By definition, for $t \in [0, h]$,

$$\frac{\mathrm{d}f(x_n(t))}{\mathrm{d}t} = \nabla f(x_n(t))^T\frac{\mathrm{d}x_n(t)}{\mathrm{d}t}$$

$$= \nabla f(x_n(t))^T v_n(t),$$

so

$$
\begin{aligned}
f(x_n(h)) &= f(x_n(0)) + \int_0^h \mathrm{d}f(x_n(t)) \\
&= f(x_n(0)) + \int_0^h \nabla f(x_n(t))^T v_n(t) \, \mathrm{d}t. \quad (13)
\end{aligned}
$$

Also, since

$$\mathrm{d}v_n(t) = (-2v_n(t) - u\nabla f(x_n(t))) \, \mathrm{d}t + 2\sqrt{u} \, \mathrm{d}B_t,$$

by Ito's lemma,

$$
\begin{aligned}
\mathrm{d}\frac{1}{2}\left\|v_n(t)\right\|^2 &= \left\langle v_n(t), 2\sqrt{u} \, \mathrm{d}B_t \right\rangle + \left( \left\langle v_n(t), -2v_n(t) - u\nabla f(x_n(t)) \right\rangle + \frac{1}{2} \cdot 4u\mathrm{Tr}(I_d) \right) \mathrm{d}t \\
&= 2\sqrt{u}v_n(t)^T \, \mathrm{d}B_t + \left( -2\left\|v_n(t)\right\|^2 - uv_n(t)^T \nabla f(x_n(t)) + 2ud \right) \mathrm{d}t,
\end{aligned}
$$

and therefore

$$\mathbb{E}\frac{1}{2u}\left\|v_n(h)\right\|^2 = \mathbb{E}\frac{1}{2u}\left\|v_n(0)\right\|^2 + \mathbb{E}\int_0^h \left( 4d - \frac{2}{u}\left\|v_n(t)\right\|^2 - v_n(t)^T \nabla f(x_n(t)) + 2d \right) \mathrm{d}t. \quad (14)$$

Now, we consider the term $\frac{1}{2u}\left\|v_n(h)\right\|^2 + f(x_n(h))$. By (13) and (14),

$$
\begin{aligned}
&\mathbb{E}\left[ \frac{1}{2u}\left\|v_n(h)\right\|^2 + f(x_n(h)) \right] \\
&= \mathbb{E}\left[ \frac{1}{2u}\left\|v_n(0)\right\|^2 + f(x_n(0)) + \int_0^h \left( -\frac{2}{u}\left\|v_n(t)\right\|^2 + 6d \right) \mathrm{d}t \right] \\
&\leq \mathbb{E}\left[ \frac{1}{2u}\left\|v_n(0)\right\|^2 + f(x_n(0)) - \frac{2}{u}h \inf_{t \in [0,h]} \left\|v_n(t)\right\|^2 + 6dh \right] \\
&\leq \mathbb{E}\left[ \frac{1}{2u}\left\|v_n(0)\right\|^2 + f(x_n(0)) \right] - \frac{2}{3}hL\mathbb{E}\left\|v_n(0)\right\|^2 + O\left( uh^3 \mathbb{E}\left\|\nabla f(x_n(0))\right\|^2 + dh \right),
\end{aligned}
$$

where the first step follows by (13) and (14) and the third step follows by Lemma 6.

Since

$$
\begin{aligned}
&\mathbb{E}\left[ \left\|v_{n+1}(0)\right\|^2 - \left\|v_n(h)\right\|^2 \right] \\
&= \mathbb{E}\left(v_{n+1}(0) - v_n(h)\right)^T \left(v_{n+1}(0) + v_n(h)\right) \\
&\leq \frac{1}{h^2}\mathbb{E}\left\|v_{n+1}(0) - v_n(h)\right\|^2 + \frac{1}{2}h^2\mathbb{E}\left\|v_{n+1}(0) + v_n(h)\right\|^2 \\
&\leq \frac{1}{h^2}\mathbb{E}\left\|v_{n+1}(0) - v_n(h)\right\|^2 + h^2\mathbb{E}\left\|v_{n+1}(0) - v_n(h)\right\|^2 + 4h^2\mathbb{E}\left\|v_n(h)\right\|^2 \\
&\leq \frac{2}{h^2}\mathbb{E}\left\|v_{n+1}(0) - v_n(h)\right\|^2 + 4h^2\mathbb{E}\left\|v_n(h)\right\|^2 \\
&\leq O\left( h^2\mathbb{E}\left\|v_n(0)\right\|^2 + u^2h^2\mathbb{E}\left\|\nabla f(x_n(0))\right\|^2 + udh^3 \right),
\end{aligned}
$$

where the first inequality follows by the inequality $2ab \leq a^2 + b^2$, the second inequality follows by Young's inequality and the last inequality follows by Lemma 2 and Lemma 6.

Since

$$\mathbb{E}f(x_{n+1}(0)) - f(x_n(h)) \leq O\left( uh^3\mathbb{E}\left\|\nabla f(x_n(0))\right\|^2 + Lh^5\mathbb{E}\left\|v_n(0)\right\|^2 + dh^6 \right),$$

which is shown in Lemma 10, we have

$$\mathbb{E}\left[ \frac{1}{2u}\left\|v_{n+1}(0)\right\|^2 + f(x_{n+1}(0)) \right]$$

$$\leq \quad \mathbb{E}\left[\frac{1}{2u}\left\|v_n(0)\right\|^2 + f(x_n(0))\right] - \frac{2}{3}hL\mathbb{E}\left\|v_n(0)\right\|^2 + O\left(uh^3\mathbb{E}\left\|\nabla f(x_n(0))\right\|^2 + dh\right)$$

$$+O\left(h^2L\mathbb{E}\left\|v_n(0)\right\|^2 + uh^2\mathbb{E}\left\|\nabla f(x_n(0))\right\|^2 + dh^3\right)$$

$$+O\left(uh^3\mathbb{E}\left\|\nabla f(x_n(0))\right\|^2 + Lh^5\mathbb{E}\left\|v_n(0)\right\|^2 + dh^6\right)$$

$$\leq \quad \mathbb{E}\left[\frac{1}{2u}\left\|v_n(0)\right\|^2 + f(x_n(0))\right] - \frac{1}{3}hL\mathbb{E}\left\|v_n(0)\right\|^2 + O\left(uh^2\mathbb{E}\left\|\nabla f(x_n(0))\right\|^2 + hd\right),$$

where the last step follows by $h$ is small. Summing $n$ from 0 to $N-1$, we get

$$\sum_{n=0}^{N-1}\mathbb{E}\left[\frac{1}{2u}\left\|v_{n+1}(0)\right\|^2 + f(x_{n+1}(0))\right]$$

$$\leq \quad \sum_{n=0}^{N-1}\mathbb{E}\left[\frac{1}{2u}\left\|v_n(0)\right\|^2 + f(x_n(0))\right] - \frac{1}{3}hL\sum_{n=0}^{N-1}\mathbb{E}\left\|v_n(0)\right\|^2$$

$$+O\left(uh^2\sum_{n=0}^{N-1}\mathbb{E}\left\|\nabla f(x_n(0))\right\|^2 + Nhd\right).$$

Since $\|v_0(0)\| = 0$ and $f(x_0(0)) \leq f(x_N(0))$,

$$\frac{1}{3}hL\sum_{n=0}^{N-1}\mathbb{E}\left\|v_n(0)\right\|^2 \quad \leq \quad O\left(uh^2\sum_{n=0}^{N-1}\mathbb{E}\left\|\nabla f(x_n(0))\right\|^2 + Nhd\right),$$

which implies

$$\sum_{n=0}^{N-1}\mathbb{E}\left\|v_n(0)\right\|^2 \quad \leq \quad O\left(u^2h\sum_{n=0}^{N-1}\mathbb{E}\left\|\nabla f(x_n(0))\right\|^2 + Nud\right).$$

$\square$

**Lemma 12.** *Assume $h$ is smaller than some given constant. For each iteration $n = 0, ..., N-1$, let $(v_n, x_n)$ be the starting point of Algorithm 1 in iteration $n$. Then, the $x_n$ in iteration $n = 0, ..., N-1$ satisfies*

$$\sum_{n=0}^{N-1}\mathbb{E}\left\|\nabla f(x_n)\right\|^2 \quad \leq \quad O\left(NLd + \frac{L}{h}\left|\mathbb{E}\nabla f(x_N)^T v_N\right|\right).$$

*Furthermore, the $v_n$ in iteration $n = 0, ..., N-1$ satisfies*

$$\sum_{n=0}^{N-1}\mathbb{E}\left\|v_n\right\|^2 \quad \leq \quad O\left(Nud + u\left|\mathbb{E}\nabla f(x_N)^T v_N\right|\right).$$

*Proof.* For each iteration $n = 0, ..., N-1$, let $\{v_n(t), x_n(t)\}_{t\in[0,h]}$ be the exact underdamped Langevin diffusion starting from $(v_n, x_n)$ computed in Algorithm 1. By definition,

$$\mathbb{E}\left[\mathrm{d}\nabla f(x_n(t))^T v_n(t)\right]$$

$$= \quad \mathbb{E}\left[v_n(t)^T\nabla^2 f(x_n(t))v_n(t) + \nabla f(x_n(t))^T \mathrm{d}v_n(t)\right]$$

$$= \quad \mathbb{E}\left[v_n(t)^T\nabla^2 f(x_n(t))v_n(t) - 2\nabla f(x_n(t))^T v_n(t) - u\left\|\nabla f(x_n(t))\right\|^2\right].$$

So we have

$$\mathbb{E}\left[\nabla f(x_n(h))^T v_n(h)\right]$$

$$= \quad \mathbb{E}\left[\nabla f(x_n(0))^T v_n(0) + \int_0^h \mathrm{d}\nabla f(x_n(t))^T v_n(t)\right]$$

$$
\begin{aligned}
&= \mathbb{E}\left[\nabla f(x_n(0))^T v_n(0) + \int_0^h v_n(t)^T \nabla^2 f(x_n(t)) v_n(t) - 2\nabla f(x_n(t))^T v_n(t) \right.\\
&\qquad\left. -u \left\|\nabla f(x_n(t))\right\|^2 \mathrm{d}t \right]\\[4pt]
&\leq \mathbb{E}\left[\nabla f(x_n(0))^T v_n(0) + 3L \int_0^h \left\|v_n(t)\right\|^2 \mathrm{d}t - \frac{1}{2}\int_0^h u \left\|\nabla f(x_n(t))\right\|^2 \mathrm{d}t \right]\\[4pt]
&\leq \mathbb{E}\left[\nabla f(x_n(0))^T v_n(0) + 3Lh \sup_{t\in[0,h]} \left\|v_n(t)\right\|^2 - \frac{1}{2}hu \inf_{t\in[0,h]} \left\|\nabla f(x_n(t))\right\|^2 \right]\\[4pt]
&\leq \mathbb{E}\nabla f(x_n(0))^T v_n(0) - \frac{1}{6}hu\mathbb{E}\left\|\nabla f(x_n(0))\right\|^2 + O\left(h^3 L\mathbb{E}\left\|v_n(0)\right\|^2 + dh^4 \right)\\[4pt]
&\quad +3Lh \cdot O\left(\mathbb{E}\left\|v_n(0)\right\|^2 + u^2 h^2 \mathbb{E}\left\|\nabla f(x_n(0))\right\|^2 + udh \right)\\[4pt]
&\leq \mathbb{E}\nabla f(x_n(0))^T v_n(0) - \frac{1}{6}hu\mathbb{E}\left\|\nabla f(x_n(0))\right\|^2\\[4pt]
&\quad +O\left(Lh\mathbb{E}\left\|v_n(0)\right\|^2 + uh^3 \mathbb{E}\left\|\nabla f(x_n(0))\right\|^2 + dh^2 \right), \qquad (15)
\end{aligned}
$$

where the third step follows by Young's inequality, the fifth step follows by Lemma 6 and the last step follows by $h$ is small. Also, we have

$$
\begin{aligned}
&\mathbb{E}\left[\nabla f(x_{n+1}(0))^T v_{n+1}(0) - \nabla f(x_n(h))^T v_n(h)\right]\\[4pt]
&= \mathbb{E}\left(\nabla f(x_{n+1}(0)) - \nabla f(x_n(h)) + \nabla f(x_n(h))\right)^T \left(v_{n+1}(0) - v_n(h)\right)\\[4pt]
&\quad +\mathbb{E}\left(\nabla f(x_{n+1}(0)) - \nabla f(x_n(h))\right)^T v_n(h)\\[4pt]
&\leq u\mathbb{E}\left\|\nabla f(x_{n+1}(0)) - \nabla f(x_n(h))\right\|^2 + L\mathbb{E}\left\|v_{n+1}(0) - v_n(h)\right\|^2 + uh^2\mathbb{E}\left\|\nabla f(x_n(h))\right\|^2\\[4pt]
&\quad +\frac{L}{h^2}\mathbb{E}\left\|v_{n+1}(0) - v_n(h)\right\|^2 + \frac{u}{h}\mathbb{E}\left\|\nabla f(x_{n+1}(0)) - \nabla f(x_n(h))\right\|^2 + hL\mathbb{E}\left\|v_n(h)\right\|^2\\[4pt]
&\leq \frac{2u}{h}\mathbb{E}\left\|\nabla f(x_{n+1}(0)) - \nabla f(x_n(h))\right\|^2 + \frac{2L}{h^2}\mathbb{E}\left\|v_{n+1}(0) - v_n(h)\right\|^2 + uh^2\mathbb{E}\left\|\nabla f(x_n(h))\right\|^2\\[4pt]
&\quad +hL\mathbb{E}\left\|v_n(h)\right\|^2\\[4pt]
&\leq \frac{2L}{h} \cdot O\left(h^6\mathbb{E}\left\|v_n(0)\right\|^2 + h^4 u^2 \mathbb{E}\left\|\nabla f(x_n(0))\right\|^2 + udh^7 \right)\\[4pt]
&\quad +\frac{2L}{h^2} \cdot O\left(h^4 \mathbb{E}\left\|v_n(0)\right\|^2 + u^2 h^4 \mathbb{E}\left\|\nabla f(x_n(0))\right\|^2 + udh^5 \right)\\[4pt]
&\quad +uh^2 \cdot O\left(\mathbb{E}\left\|\nabla f(x_n(0))\right\|^2 + L^2 h^2 \mathbb{E}\left\|v_n(0)\right\|^2 + Ldh^3 \right)\\[4pt]
&\quad +hL \cdot O\left(\mathbb{E}\left\|v_n(0)\right\|^2 + u^2 h^2 \mathbb{E}\left\|\nabla f(x_n(0))\right\|^2 + udh \right)\\[4pt]
&\leq O\left(hL\mathbb{E}\left\|v_n(0)\right\|^2 + uh^2\mathbb{E}\left\|\nabla f(x_n(0))\right\|^2 + dh^2 \right), \qquad (16)
\end{aligned}
$$

where the second step follows by Young's inequality and the fourth step follows by Lemma 2 and Lemma 6. Combining (15) and (16),

$$
\begin{aligned}
\mathbb{E}\nabla f(x_{n+1}(0))^T v_{n+1}(0) &\leq \mathbb{E}\nabla f(x_n(0))^T v_n(0) - \frac{1}{6}hu\mathbb{E}\left\|\nabla f(x_n(0))\right\|^2\\[4pt]
&\quad +O\left(Lh\mathbb{E}\left\|v_n(0)\right\|^2 + uh^3\mathbb{E}\left\|\nabla f(x_n(0))\right\|^2 + dh^2 \right)\\[4pt]
&\quad +O\left(Lh\mathbb{E}\left\|v_n(0)\right\|^2 + uh^2\mathbb{E}\left\|\nabla f(x_n(0))\right\|^2 + dh^2 \right)\\[4pt]
&\leq \mathbb{E}\nabla f(x_n(0))^T v_n(0) - \frac{1}{6}hu\mathbb{E}\left\|\nabla f(x_n(0))\right\|^2\\[4pt]
&\quad +O\left(Lh\mathbb{E}\left\|v_n(0)\right\|^2 + uh^2\mathbb{E}\left\|\nabla f(x_n(0))\right\|^2 + dh^2 \right).
\end{aligned}
$$

Summing from $n = 0$ to $N - 1$,

$$\sum_{n=0}^{N-1} \mathbb{E}\nabla f(x_{n+1}(0))^T v_{n+1}(0) \leq \sum_{n=0}^{N-1} \mathbb{E}\nabla f(x_n(0))^T v_n(0) - \frac{1}{6}hu \sum_{n=0}^{N-1} \mathbb{E}\left\|\nabla f(x_n(0))\right\|^2$$

$$+ O\left(Lh \sum_{n=0}^{N-1} \mathbb{E}\left\|v_n(0)\right\|^2 + uh^2 \sum_{n=0}^{N-1} \mathbb{E}\left\|\nabla f(x_n(0))\right\|^2 + Ndh^2\right)$$

$$\leq \sum_{n=0}^{N-1} \mathbb{E}\nabla f(x_n(0))^T v_n(0) - \frac{1}{6}hu \sum_{n=0}^{N-1} \left\|\nabla f(x_n(0))\right\|^2$$

$$+ O\left(Lh\left(u^2 h \sum_{n=0}^{N-1} \mathbb{E}\left\|\nabla f(x_n(0))\right\|^2 + Nud\right) + Ndh^2\right)$$

$$\leq \sum_{n=0}^{N-1} \mathbb{E}\nabla f(x_n(0))^T v_n(0) - \frac{1}{8}hu \sum_{n=0}^{N-1} \left\|\nabla f(x_n(0))\right\|^2 + O\left(Ndh\right),$$

where the second step follows by Lemma 11 and the last step follows by $h$ is small. Then, since $v_0 = 0$,

$$\frac{1}{8}hu \sum_{n=0}^{N-1} \mathbb{E}\left\|\nabla f(x_n(0))\right\|^2 \quad \leq \quad O\left(Ndh + \left|\mathbb{E}\nabla f(x_N(0))^T v_N(0)\right|\right),$$

which implies

$$\sum_{n=0}^{N-1} \mathbb{E}\left\|\nabla f(x_n(0))\right\|^2 \quad \leq \quad O\left(NLd + \frac{L}{h}\left|\mathbb{E}\nabla f(x_N(0))^T v_N(0)\right|\right).$$

By Lemma 11,

$$\sum_{n=0}^{N-1} \mathbb{E}\left\|v_n(0)\right\|^2 \quad \leq \quad O\left(u^2 h \sum_{n=0}^{N-1} \mathbb{E}\left\|\nabla f(x_n(0))\right\|^2 + Nud\right)$$

$$\leq \quad O\left(Nud + u\left|\mathbb{E}\nabla f(x_N(0))^T v_N(0)\right|\right).$$

$\square$

## E    Proof of Theorem 3

Here, we combine Lemma 12 and Lemma 2 to prove our main result.

*Proof.* Let $x_{n+\frac{1}{2}}$, $x_n$ and $v_n$ be the iterates of Algorithm 1. Let $(y_n, w_n)$ be the $n$-th step of the exact underdamped Langevin diffusion, starting from a random point $(y_0, w_0) \propto \exp\left(-\left(f(y) + \frac{L}{2}\left\|w\right\|^2\right)\right)$, coupled with $(x_n, v_n)$ through the same Brownian motion. Let $\left(x_{n+1}^*, v_{n+1}^*\right)$ be the 1-step exact Langevin diffusion starting from $(x_n, v_n)$. For any iteration $n$, let $\mathbb{E}_\alpha$ be the expectation taken over the random choice of $\alpha$ in iteration $n$. Then,

$$\mathbb{E}_\alpha \left[\left\|x_n - y_n\right\|^2 + \left\|(x_n + v_n) - (y_n + w_n)\right\|^2\right]$$

$$= \quad \mathbb{E}_\alpha \left[\left\|(x_n - x_n^*) - (y_n - x_n^*)\right\|^2 + \left\|(x_n + v_n - x_n^* - v_n^*) - (y_n + w_n - x_n^* - v_n^*)\right\|^2\right]$$

$$\leq \quad \left\|y_n - x_n^*\right\|^2 + \left\|y_n + w_n - x_n^* - v_n^*\right\|^2 + \mathbb{E}_\alpha \left\|x_n - x_n^*\right\|^2 + \mathbb{E}_\alpha \left\|x_n + v_n - x_n^* - v_n^*\right\|^2$$

$$-2\left(y_n - x_n^*\right)^T \left(\mathbb{E}_\alpha x_n - x_n^*\right) - 2\left(y_n + w_n - x_n^* - v_n^*\right)^T \left(\mathbb{E}_\alpha\left[x_n + v_n\right] - x_n^* - v_n^*\right)$$

$$\leq \quad \left(1 + \frac{h}{2\kappa}\right)\left(\left\|y_n - x_n^*\right\|^2 + \left\|y_n + w_n - x_n^* - v_n^*\right\|^2\right)$$

$$+\frac{2\kappa}{h}\left(\left\|\mathbb{E}_\alpha x_n - x_n^*\right\|^2 + \left\|\mathbb{E}_\alpha\left[x_n + v_n\right] - x_n^* - v_n^*\right\|^2\right) + \mathbb{E}_\alpha \left\|x_n - x_n^*\right\|^2$$

$$+\mathbb{E}_\alpha \left\| x_n + v_n - x_n^* - v_n^* \right\|^2,$$

where the second step follows by $y_n, w_n, x_n^*$ and $v_n^*$ are independent of the choice of $\alpha$ and the third follows by Young's inequality. Then,

$$\mathbb{E}\left[ \|x_N - y_N\|^2 + \|(x_N + v_N) - (y_N + w_N)\|^2 \right]$$

$$\leq \left(1 + \frac{h}{2\kappa}\right) e^{-\frac{h}{\kappa}} \mathbb{E}\left[ \|y_{N-1} - x_{N-1}\|^2 + \|y_{N-1} + w_{N-1} - x_{N-1} - v_{N-1}\|^2 \right]$$

$$+ \frac{2\kappa}{h} \left( \mathbb{E}\|\mathbb{E}_\alpha x_N - x_N^*\|^2 + \mathbb{E}\|\mathbb{E}_\alpha x_N + v_N - x_N^* - v_N^*\|^2 \right)$$

$$+ \left( \mathbb{E}\|x_N - x_N^*\|^2 + \mathbb{E}\|x_N + v_N - x_N^* - v_N^*\|^2 \right)$$

$$\leq e^{-\frac{h}{2\kappa}} \mathbb{E}\left[ \|y_{N-1} - x_{N-1}\|^2 + \|y_{N-1} + w_{N-1} - x_{N-1} - v_{N-1}\|^2 \right]$$

$$+ \frac{2\kappa}{h} \left( 2\mathbb{E}\|\mathbb{E}_\alpha v_N - v_N^*\|^2 + 3\mathbb{E}\|\mathbb{E}_\alpha x_N - x_N^*\|^2 \right) + \left( 2\mathbb{E}\|v_N - v_N^*\|^2 + 3\mathbb{E}\|x_N - x_N^*\|^2 \right)$$

$$\leq e^{-\frac{Nh}{2\kappa}} \mathbb{E}\left[ \|y_0 - x_0\|^2 + \|y_0 + w_0 - x_0 - v_0\|^2 \right]$$

$$+ \sum_{n=1}^{N} \frac{2\kappa}{h} \left( 2\mathbb{E}\|\mathbb{E}_\alpha v_n - v_n^*\|^2 + 3\mathbb{E}\|\mathbb{E}_\alpha x_n - x_n^*\|^2 \right)$$

$$+ \sum_{n=1}^{N} \left( 2\mathbb{E}\|v_n - v_n^*\|^2 + 3\mathbb{E}\|x_n - x_n^*\|^2 \right),$$

where the first step follows by Lemma 1, the second step follows by $1 + \frac{h}{2\kappa} \leq e^{\frac{h}{2\kappa}}$, and the last step follows by induction.

Since $(y_N, w_N)$ follows the distribution $p^* \propto \exp\left( -\left( f(y) + \frac{L}{2}\|w\|^2 \right) \right)$, $\mathbb{E}\|w_N\|^2 = \frac{d}{L}$. By Proposition 1 of [19], $\mathbb{E}\|y_0 - x_0\|^2 \leq \frac{d}{m}$. Then,

$$\mathbb{E}\left[ \|y_0 - x_0\|^2 + \|y_0 + w_0 - x_0 - v_0\|^2 \right] \leq 3\mathbb{E}\|y_0 - x_0\|^2 + 2\mathbb{E}\|w_0 - v_0\|^2$$

$$\leq 5\frac{d}{m}.$$

When $N = \frac{2\kappa}{h} \log\left(\frac{20}{\epsilon^2}\right)$,

$$e^{-\frac{Nh}{2\kappa}} \mathbb{E}\left[ \|y_0 - x_0\|^2 + \|y_0 + w_0 - x_0 - v_0\|^2 \right] \leq \frac{\epsilon^2 d}{4m}.$$

By Lemma 2,

$$\sum_{n=1}^{N} \frac{2\kappa}{h} \left( 2\mathbb{E}\|\mathbb{E}_\alpha v_n - v_n^*\|^2 + 3\mathbb{E}\|\mathbb{E}_\alpha x_n - x_n^*\|^2 \right)$$

$$\leq O\left( h^7 \kappa \sum_{n=0}^{N-1} \mathbb{E}\|v_n\|^2 + \frac{u}{m} h^9 \sum_{n=0}^{N-1} \mathbb{E}\|\nabla f(x_n)\|^2 + \frac{1}{m} N d h^8 \right),$$

and

$$\sum_{n=1}^{N} \left( 2\mathbb{E}\|v_n - v_n^*\|^2 + 3\mathbb{E}\|x_n - x_n^*\|^2 \right)$$

$$\leq O\left( h^4 \sum_{n=0}^{N-1} \mathbb{E}\|v_n\|^2 + u^2 h^4 \sum_{n=0}^{N-1} \mathbb{E}\|\nabla f(x_n)\|^2 + N u d h^5 \right).$$

By Lemma 2 of [12], $\mathbb{E}\|\nabla f(y_N)\|^2 \leq dL$. Then, by $\mathbb{E}\|\nabla f(y_N)\|^2 \leq dL$ and $\mathbb{E}\|w_N\|^2 = \frac{d}{L}$,

$$\left| \mathbb{E}\nabla f(x_N)^T v_N \right| \leq \mathbb{E}\left[ L\|v_N\|^2 + u\|\nabla f(x_N)\|^2 \right]$$

$$\leq \quad 2\mathbb{E}\left[L\|w_N\|^2 + L\|v_N - w_N\|^2 + u\|\nabla f(y_N)\|^2 + L\|x_N - y_N\|^2\right]$$

$$\leq \quad 4d + 2L\mathbb{E}\left[\|v_N - w_N\|^2 + \|x_N - y_N\|^2\right]$$

$$\leq \quad 4d + 6L\mathbb{E}\left[\|x_N - y_N\|^2 + \|(x_N + v_N) - (y_N + w_N)\|^2\right],$$

By Lemma 12 and our choice of $N$,

$$\sum_{n=0}^{N-1} \|\nabla f(x_n(0))\|^2 \leq O\left(\frac{\kappa dL}{h}\log\left(\frac{1}{\epsilon^2}\right) + \frac{L^2}{h}\mathbb{E}\left[\|x_N - y_N\|^2 + \|(x_N + v_N) - (y_N + w_N)\|^2\right]\right),$$

and

$$\sum_{n=0}^{N-1} \mathbb{E}\|v_n(0)\|^2 \leq O\left(\frac{d}{hm}\log\left(\frac{1}{\epsilon^2}\right) + \mathbb{E}\left[\|x_N - y_N\|^2 + \|(x_N + v_N) - (y_N + w_N)\|^2\right]\right).$$

Thus,

$$\sum_{n=1}^{N} \frac{2\kappa}{h}\left(2\mathbb{E}\|\mathbb{E}_\alpha v_n - v_n^*\|^2 + 3\mathbb{E}\|\mathbb{E}_\alpha x_n - x_n^*\|^2\right) + \sum_{n=1}^{N}\left(2\mathbb{E}\|v_n - v_n^*\|^2 + 3\mathbb{E}\|x_n - x_n^*\|^2\right)$$

$$\leq O\left(\left(\frac{\kappa dh^6}{m} + \frac{dh^3}{m}\right)\log\left(\frac{1}{\epsilon^2}\right)\right)$$

$$+ O\left(\kappa h^7 + h^3\right)\mathbb{E}\left[\|x_N - y_N\|^2 + \|(x_N + v_N) - (y_N + w_N)\|^2\right].$$

Then, we can choose a small constant $C$ such that if we let

$$h = C\min\left(\frac{\epsilon^{1/3}}{\kappa^{1/6}}\log^{-1/6}\left(\frac{1}{\epsilon^2}\right), \epsilon^{2/3}\log^{-1/3}\left(\frac{1}{\epsilon^2}\right)\right),$$

then

$$\sum_{n=1}^{N} \frac{2\kappa}{h}\left(2\mathbb{E}\|\mathbb{E}_\alpha v_n - v_n^*\|^2 + 3\mathbb{E}\|\mathbb{E}_\alpha x_n - x_n^*\|^2\right) + \sum_{n=1}^{N}\left(2\mathbb{E}\|v_n - v_n^*\|^2 + 3\mathbb{E}\|x_n - x_n^*\|^2\right)$$

$$\leq \frac{\epsilon^2 d}{4m} + \frac{1}{2}\mathbb{E}\left[\|x_N - y_N\|^2 + \|(x_N + v_N) - (y_N + w_N)\|^2\right].$$

Therefore,

$$\mathbb{E}\left[\|x_N - y_N\|^2 + \|(x_N + v_N) - (y_N + w_N)\|^2\right]$$

$$\leq \quad \frac{\epsilon^2 d}{4m} + \frac{\epsilon^2 d}{4m} + \frac{1}{2}\mathbb{E}\left[\|x_N - y_N\|^2 + \|(x_N + v_N) - (y_N + w_N)\|^2\right]$$

$$= \quad \frac{\epsilon^2 d}{2m} + \frac{1}{2}\mathbb{E}\left[\|x_N - y_N\|^2 + \|(x_N + v_N) - (y_N + w_N)\|^2\right],$$

which implies

$$\mathbb{E}\left[\|x_N - y_N\|^2\right] \leq \mathbb{E}\left[\|x_N - y_N\|^2 + \|(x_N + v_N) - (y_N + w_N)\|^2\right] \leq \frac{\epsilon^2 d}{m}.$$

By our choice of $h$,

$$N \quad \leq \quad \tilde{O}\left(\frac{\kappa^{7/6}}{\epsilon^{1/3}} + \frac{\kappa}{\epsilon^{2/3}}\right).$$

$\square$

# F   Discretization Error of Algorithm 2

Here, we bound the discretization error in one step of Algorithm 2. Since the terms $\mathbb{E}\left\|\mathbb{E}_\alpha x_{n+1} - x_n^*(h)\right\|^2$ and $\mathbb{E}\left\|x_{n+1} - x_n^*(h)\right\|^2$ are dominated by the terms $\mathbb{E}\left\|\mathbb{E}_\alpha v_{n+1} - v_n^*(h)\right\|^2$ and $\mathbb{E}\left\|v_{n+1} - v_n^*(h)\right\|^2$, we bound only the later two terms.

**Lemma 13.** *Assume that $R^4\delta^4 \leq \frac{1}{4}$. Let $x_n^{(k-1,i)}$ for $i = 1, ..., R$, $k = 1, ..., K$ be the intermediate value computed in iteration $n$ of Algorithm 2. Let $\{x_n^*(t), v_n^*(t)\}_{t \in [0,h]}$ be the ideal underdamped Langevin diffusion, starting from $x_n^*(0) = x_n$ and $v_n^*(0) = v_n$, coupled through a shared Brownian motion with $\left\{x_n^{(k-1,i)}\right\}_{i=1,...,R,k=1,...,K}$. Then, for any $i = 1, ..., R$, and $k = 1, ..., K - 1$,*

$$
\mathbb{E}\left\|x_n^{(k,i)} - x_n^*(\alpha_i h)\right\|^2 \leq \left(2R^4\delta^4\right)^k \frac{1}{R}\sum_{j=1}^{R} \mathbb{E}\left\|x_n - x_n^*(\alpha_j h)\right\|^2
$$

$$
+ 4R^3\delta^4 \sum_{j=1}^{R} \mathbb{E}\sup_{s \in [(j-1)\delta, j\delta]}\left\|x_n^*(\alpha_j h) - x_n^*(s)\right\|^2.
$$

*Proof.* For any $i = 1, ..., R$, and $k = 1, ..., K - 1$,

$$
\mathbb{E}\left\|x_n^{(k,i)} - x_n^*(\alpha_i h)\right\|^2
$$

$$
\leq \quad \mathbb{E}\left\|\frac{1}{2}u\sum_{j=1}^{i}\left[\int_{(j-1)\delta}^{\min(j\delta,\alpha_i h)}\left(1 - e^{-2(\alpha_i h - s)}\right)\mathrm{d}s \cdot \nabla f(x_n^{(k-1,j)})\right] \right.
$$

$$
\left. -\frac{1}{2}u\int_0^{\alpha_i h}\left(1 - e^{-2(\alpha_i h - s)}\right)\nabla f(x_n^*(s))\,\mathrm{d}s\right\|^2
$$

$$
\leq \quad \frac{1}{2}\mathbb{E}\left\|u\sum_{j=1}^{i}\left[\int_{(j-1)\delta}^{\min(j\delta,\alpha_i h)}\left(1 - e^{-2(\alpha_i h - s)}\right)\mathrm{d}s \cdot \left(\nabla f(x_n^{(k-1,j)}) - \nabla f(x_n^*(\alpha_j h))\right)\right]\right\|^2
$$

$$
+ \frac{1}{2}\mathbb{E}\left\|u\sum_{j=1}^{i}\left[\int_{(j-1)\delta}^{\min(j\delta,\alpha_i h)}\left(1 - e^{-2(\alpha_i h - s)}\right)\left(\nabla f(x_n^*(\alpha_j h)) - \nabla f(x_n^*(s))\right)\mathrm{d}s\right]\right\|^2,
$$

where the first step follows by the definition, and the second step follows by Young's inequality.

To compute the first term,

$$
\frac{1}{2}\mathbb{E}\left\|u\sum_{j=1}^{i}\left[\int_{(j-1)\delta}^{\min(j\delta,\alpha_i h)}\left(1 - e^{-2(\alpha_i h - s)}\right)\mathrm{d}s \cdot \left(\nabla f(x_n^{(k-1,j)}) - \nabla f(x_n^*(\alpha_j h))\right)\right]\right\|^2
$$

$$
\leq \quad \frac{1}{2}u^2 R\sum_{j=1}^{i}\mathbb{E}\left\|\int_{(j-1)\delta}^{\min(j\delta,\alpha_i h)}\left(1 - e^{-2(\alpha_i h - s)}\right)\mathrm{d}s \cdot \left(\nabla f(x_n^{(k-1,j)}) - \nabla f(x_n^*(\alpha_j h))\right)\right\|^2
$$

$$
\leq \quad 2R^3\delta^4 \sum_{j=1}^{R}\mathbb{E}\left\|x_n^{(k-1,j)} - x_n^*(\alpha_j h)\right\|^2, \tag{17}
$$

where the first step follows by the inequality $\left(\sum_{i=1}^{n} a_i\right)^2 \leq n\sum_{i=1}^{n} a_i^2$, the second step follows by $1 - e^{-2(\alpha_i h - s)} \leq 2R\delta$ and $\nabla f$ is $L$-Lipschitz.

For the second term,

$$
\frac{1}{2}\mathbb{E}\left\|u\sum_{j=1}^{i}\left[\int_{(j-1)\delta}^{\min(j\delta,\alpha_i h)}\left(1 - e^{-2(\alpha_i h - s)}\right)\left(\nabla f(x_n^*(\alpha_j h)) - \nabla f(x_n^*(s))\right)\mathrm{d}s\right]\right\|^2
$$

$$\leq \quad \frac{1}{2}u^2 R \sum_{j=1}^{i} \mathbb{E} \left\| \int_{(j-1)\delta}^{\min(j\delta, \alpha_i h)} \left( 1 - e^{-2(\alpha_i h - s)} \right) (\nabla f(x_n^*(\alpha_j h)) - \nabla f(x_n^*(s))) \, \mathrm{d}s \right\|^2$$

$$\leq \quad 2R^3 \delta^4 \sum_{j=1}^{R} \mathbb{E} \sup_{s \in [(j-1)\delta, j\delta]} \|x_n^*(\alpha_j h) - x_n^*(s)\|^2, \tag{18}$$

where the first step follows by the inequality $\left( \sum_{i=1}^{n} a_i \right)^2 \leq n \sum_{i=1}^{n} a_i^2$ and the second step follows by $1 - e^{-2(\alpha_i h - s)} \leq 2R\delta$ and $\nabla f$ is $L$-Lipschitz. Thus,

$$\mathbb{E} \left\| x_n^{(k,i)} - x_n^*(\alpha_i h) \right\|^2$$

$$\leq \quad 2R^3 \delta^4 \sum_{j=1}^{R} \mathbb{E} \left\| x_n^{(k-1,j)} - x_n^*(\alpha_j h) \right\|^2 + 2R^3 \delta^4 \sum_{j=1}^{R} \mathbb{E} \sup_{s \in [(j-1)\delta, j\delta]} \|x_n^*(\alpha_j h) - x_n^*(s)\|^2$$

$$\leq \quad \left( 2R^4 \delta^4 \right)^k \frac{1}{R} \sum_{j=1}^{R} \mathbb{E} \|x_n - x_n^*(\alpha_j h)\|^2$$

$$+ \left( 1 + 2R^4 \delta^4 + ... + \left( 2R^4 \delta^4 \right)^{k-1} \right) 2R^3 \delta^4 \sum_{j=1}^{R} \mathbb{E} \sup_{s \in [(j-1)\delta, j\delta]} \|x_n^*(\alpha_j h) - x_n^*(s)\|^2$$

$$\leq \quad \left( 2R^4 \delta^4 \right)^k \frac{1}{R} \sum_{j=1}^{R} \mathbb{E} \|x_n - x_n^*(\alpha_j h)\|^2 + 4R^3 \delta^4 \sum_{j=1}^{R} \mathbb{E} \sup_{s \in [(j-1)\delta, j\delta]} \|x_n^*(\alpha_j h) - x_n^*(s)\|^2,$$

where the first step follows by (17) and (18), the second step follows by induction, and the third step follows by $2R^4 \delta^4 \leq \frac{1}{2}$. □

**Lemma 14.** *Let $(v_n, x_n)$ be the iterates of iteration $n$. Let $x_n^{(k,i)}$ for $i = 1, ..., R$, $k = 1, ..., K-1$ be the intermediate value computed in iteration $n$ of Algorithm 2. Let $\{x_n^*(t), v_n^*(t)\}_{t \in [0,h]}$ be the ideal underdamped Langevin diffusion, starting from $x_n^*(0) = x_n$ and $v_n^*(0) = v_n$, coupled through a shared Brownian motion with $\left\{ x_n^{(k,i)} \right\}_{i=1,...,R, k=1,...,K-1}$. Assume that $h = R\delta \leq \frac{1}{10}$ and $K \geq \Omega \left( \log \frac{1}{\delta^4} \right)$. Let $\mathbb{E}_\alpha$ be the expectation taken over the choice of $\alpha_1, ..., \alpha_R$ in iteration $n$. Let $\mathbb{E}$ be the expectation taken over other randomness in iteration $n$. Then,*

$$\mathbb{E} \|\mathbb{E}_\alpha v_{n+1} - v_n^*(h)\|^2 \leq O \left( R^6 \delta^8 \|v_n\|^2 + u^2 R^6 \delta^{10} \|\nabla f(x_n)\|^2 + R^6 \delta^9 ud \right),$$

$$\mathbb{E} \|v_{n+1} - v_n^*(h)\|^2 \leq O \left( R^2 \delta^4 \|v_n\|^2 + u^2 R^2 \delta^4 \|\nabla f(x_n)\|^2 + R^2 \delta^5 ud \right).$$

*Proof.* To show the first claim,

$$\mathbb{E} \|\mathbb{E}_\alpha v_{n+1} - v_n^*(h)\|^2$$

$$\leq \quad \mathbb{E} \left\| \mathbb{E}_\alpha u \sum_{i=1}^{R} \delta e^{-2(h - \alpha_i h)} \nabla f(x_n^{(K-1,i)}) - u \int_0^h e^{-2(h-s)} \nabla f(x_n^*(s)) \, \mathrm{d}s \right\|^2$$

$$\leq \quad 2\mathbb{E} \left\| u \sum_{i=1}^{R} \delta e^{-2(h - \alpha_i h)} \nabla f(x_n^{(K-1,i)}) - u \sum_{i=1}^{R} \delta e^{-2(h - \alpha_i h)} \nabla f(x_n^*(\alpha_i h)) \right\|^2$$

$$+ 2\mathbb{E} \left\| \mathbb{E}_\alpha u \sum_{i=1}^{R} \delta e^{-2(h - \alpha_i h)} \nabla f(x_n^*(\alpha_i h)) - u \int_0^h e^{-2(h-s)} \nabla f(x_n^*(s)) \, \mathrm{d}s \right\|^2$$

$$\leq \quad 2\delta^2 R \sum_{i=1}^{R} \mathbb{E} \left\| x_n^{(K-1,i)} - x_n^*(\alpha_i h) \right\|^2 + 0$$

$$\leq \quad 2\delta^2 R \left( 2R^4 \delta^4 \right)^{K-1} \sum_{i=1}^{R} \mathbb{E} \|x_n - x_n^*(\alpha_i h)\|^2$$

$$+ 8R^5\delta^6 \sum_{i=1}^{R} \mathbb{E} \sup_{s \in [(i-1)\delta, i\delta]} \|x_n^*(\alpha_i h) - x_n^*(s)\|^2, \tag{19}$$

where the first step follows by the definition, the second step follows by Young's inequality, and the third step follows by

$$\mathbb{E}_\alpha \delta e^{-2(h-\alpha_i h)} \nabla f(x_n^*(\alpha_i h)) = \int_{(i-1)\delta}^{i\delta} e^{-2(h-s)} \nabla f(x_n^*(s)) \, \mathrm{d}s.$$

To show the second claim,

$$\mathbb{E} \|v_{n+1} - v_n^*(h)\|^2$$

$$\leq \mathbb{E} \left\| u \sum_{i=1}^{R} \delta e^{-2(h-\alpha_i h)} \nabla f(x_n^{(K-1,i)}) - u \int_0^h e^{-2(h-s)} \nabla f(x_n^*(s)) \, \mathrm{d}s \right\|^2$$

$$\leq 3\mathbb{E} \left\| u \sum_{i=1}^{R} \delta e^{-2(h-\alpha_i h)} \nabla f(x_n^{(K-1,i)}) - u \sum_{i=1}^{R} \delta e^{-2(h-\alpha_i h)} \nabla f(x_n^*(\alpha_i h)) \right\|^2$$

$$+ 3\mathbb{E} \left\| u \sum_{i=1}^{R} \int_{(i-1)\delta}^{i\delta} e^{-2(h-\alpha_i h)} \left( \nabla f(x_n^*(\alpha_i h)) - \nabla f(x_n^*(s)) \right) \, \mathrm{d}s \right\|^2$$

$$+ 3\mathbb{E} \left\| u \sum_{i=1}^{R} \int_{(i-1)\delta}^{i\delta} \left( e^{-2(h-\alpha_i h)} - e^{-2(h-s)} \right) \nabla f(x_n^*(s)) \, \mathrm{d}s \right\|^2.$$

Like the proof of the third claim, the first term satisfies

$$3\mathbb{E} \left\| u \sum_{i=1}^{R} \delta e^{-2(h-\alpha_i h)} \nabla f(x_n^{(K-1,i)}) - u \sum_{i=1}^{R} \delta e^{-2(h-\alpha_i h)} \nabla f(x_n^*(\alpha_i h)) \right\|^2$$

$$\leq 3\delta^2 R \left( 2R^4\delta^4 \right)^{K-1} \sum_{i=1}^{R} \mathbb{E} \|x_n - x_n^*(\alpha_i h)\|^2 + 12R^5\delta^6 \sum_{i=1}^{R} \mathbb{E} \sup_{s \in [(i-1)\delta, i\delta]} \|x_n^*(\alpha_i h) - x_n^*(s)\|^2.$$

The second term satisfies

$$3\mathbb{E} \left\| u \sum_{i=1}^{R} \int_{(i-1)\delta}^{i\delta} e^{-2(h-\alpha_i h)} \left( \nabla f(x_n^*(\alpha_i h)) - \nabla f(x_n^*(s)) \right) \, \mathrm{d}s \right\|^2$$

$$\leq 3u^2 R \sum_{i=1}^{R} \mathbb{E} \left\| \int_{(i-1)\delta}^{i\delta} e^{-2(h-\alpha_i h)} \left( \nabla f(x_n^*(\alpha_i h)) - \nabla f(x_n^*(s)) \right) \, \mathrm{d}s \right\|^2$$

$$\leq 3\delta^2 R \sum_{i=1}^{R} \mathbb{E} \sup_{s \in [(i-1)\delta, i\delta]} \|x_n^*(\alpha_i h) - x_n^*(s)\|^2,$$

where the first step follows by $\left( \sum_{i=1}^{n} a_i \right)^2 \leq n \sum_{i=1}^{n} a_i^2$, and the second step follows by $\nabla f$ is $L$-Lipschitz.

The last term satisfies

$$3\mathbb{E} \left\| u \sum_{i=1}^{R} \int_{(i-1)\delta}^{i\delta} \left( e^{-2(h-\alpha_i h)} - e^{-2(h-s)} \right) \nabla f(x_n^*(s)) \, \mathrm{d}s \right\|^2 \leq 12u^2 R^2 \delta^4 \mathbb{E} \sup_{s \in [0,h]} \|\nabla f(x_n^*(s))\|^2,$$

which follows by $e^{-2(h-\alpha_i h)} - e^{-2(h-s)} \leq 2\delta$ for $s \in [(i-1)\delta, i\delta]$. Thus,

$$\mathbb{E} \|v_{n+1} - v_n^*(h)\|^2$$

$$\leq 3\delta^2 R \left( 2R^4\delta^4 \right)^{K-1} \sum_{i=1}^{R} \mathbb{E} \|x_n - x_n^*(\alpha_i h)\|^2$$

$$+12R^5\delta^6\sum_{i=1}^{R}\mathbb{E}\sup_{s\in[(i-1)\delta,i\delta]}\|x_n^*(\alpha_i h)-x_n^*(s)\|^2$$

$$+3\delta^2 R\sum_{i=1}^{R}\mathbb{E}\sup_{s\in[(i-1)\delta,i\delta]}\|x_n^*(\alpha_i h)-x_n^*(s)\|^2+12u^2R^2\delta^4\mathbb{E}\sup_{s\in[0,h]}\|\nabla f(x_n^*(s))\|^2 \;(20)$$

By Lemma 6, for $i=1,...,R$,

$$\mathbb{E}\|x_n-x_n^*(\alpha_i h)\|^2 \;\leq\; O\left(R^2\delta^2\|v_n\|^2+u^2R^4\delta^4\|\nabla f(x_n)\|^2+udR^3\delta^3\right),$$

,and

$$\mathbb{E}\sup_{s\in[(i-1)\delta,i\delta]}\|x_n^*(\alpha_i h)-x_n^*(s)\|^2\leq O\left(\delta^2\|v_n\|^2+u^2\delta^4\|\nabla f(x_n)\|^2+ud\delta^3\right).$$

Thus, when $K\geq\Omega\left(\log\frac{1}{\delta^4}\right)$, since $R\delta\leq\frac{1}{10}$, $\left(2R^4\delta^4\right)^{K-1}\leq O\left(\delta^4\right)$. By (19) and (20),

$$\mathbb{E}\|\mathbb{E}_\alpha v_{n+1}-v_n^*(h)\|^2 \;\leq\; O\left(R^6\delta^8\|v_n\|^2+u^2R^6\delta^{10}\|\nabla f(x_n)\|^2+R^6\delta^9 ud\right),$$

and

$$\mathbb{E}\|v_{n+1}-v_n^*(h)\|^2 \;\leq\; O\left(R^2\delta^4\|v_n\|^2+u^2R^2\delta^4\mathbb{E}\|\nabla f(x_n)\|^2+R^2\delta^5 ud\right).$$

$\square$