[Reviews · NeurIPS 2019]

Reviewer 1



SUMMARY This papers contains a new and original contribution to the rapidly increasing literature on sampling from a multivariate strongly-log-concave density function. The standard setting is considered: the potential function is m-strongly convex and L-gradient Lipschitz. The authors propose a new discretization of the underdamped Langevin diffusion, which leads to an improved approximation of the invariant density. EVALUATION I found this paper very interesting. It is based on the following very simple idea of randomization. Instead of approximating an integral \int_0^h f(x_t), for a small h, by h*f(x_0), one can approximate it by the unbiased random variable h*f(x_{h*a}), where a is a r.v. uniformly distributed in [0,1]. I am sure many researchers working on the problem of sampling have already had this idea, but to the best of my knowledge, none of them have managed to convert it into an improved approximation bound. The main high-level contribution of the paper is that it shows how one can take advantage of the aforementioned randomisation trick. The latter is quite general and can be applicable in other situations than the one described in the paper. I have checked almost 90% of proofs and find that they are correct. The paper could be better polished, but I would not penalise it for this reason. The material is clearly presented. There are two minor weaknesses, listed below, that should not be difficult to fix in the final version of the paper. - in the comparison with the prior work, the authors do not present the latest results. Therefore, the comparison embellishes the contributions of this paper. Indeed, in table 1, in the row corresponding to the Langevin diffusion, the authors should present the result of "Analysis of Langevin Monte Carlo via convex optimization", which improves on [13,17]. In addition, for the Underdamped Langevin Diffusion, in the table,in the abstract and in the text, it would be more appropriate to compare with [15], which provides an improved bound (kappa^1.5 instead of kappa^2). - The authors do not keep track of numerical constants. I would very much appreciate to see results with precise values of constants, as it is done in many recent references cited by the authors. SPECIFIC REMARKS/TYPOS - Line 9: compare with [15] instead of [10] - Line 14: applies -> apply - Line 16: problems -> problem - Lines 32-33: this lines oversell the contribution (which is not needed). If one uses the methodology of evaluating a method adopted by the authors, that I find fully justified, the current best algorithm is independent of d. By the way, the sentence on lines 32-33 contradicts the content of table 1. - Lines 44-45: when mentioning the current fastest algorithm'', it might be helpful to cite [15] as well. - Lines 106-107 - same remark - Line 131: depend -> depends - Line 132: I guess Omega(d^1.5) should be replaced by Omega(d^2) - Line 136: I suggest to use another letter than f, which refers to a precise function (the potential) - Lines 170-171: [18,15] -> [18,13] - Lines 180-181: the meaning of "accurate" is not clear here. My suggestion is to remove any comment about accuracy here and to keep only the comment on unbiasedness. - Line 191: please emphasize that the distribution is Gaussian conditionnally to alpha. - Lemma 2: it should be clearly mentioned that these claims are true when u=1/L - Lemma 2: I suggest to replace v_n(0) and x_n(0) by v_n and x_n - Line 201: same remark - Section A: everything is conditional to alpha - Lemma 6: mention that u=1/L - Line 400: Schwarz is mispelled

Reviewer 2



This work may be of considerable interest to people working on (applications of) stochastic differential equations. It is generally well written and appears to be technically sound (to the extend I could verify). What I considered to be missing in this paper was "at least an illustration of how the algorithm might eventually materialize into a machine learning application" (quoted from the NeurIPS CFP).

Reviewer 3



------------------- Updates after author feedback. I have read your feedback carefully. I really appreciate your efforts doing the simulation analysis. The plots look good. Several details like how to evaluate distributional distance epsilon numerically and how to tune the step-size is missing, but I think you will have them in the final version. I would like to maintain the accept suggestion. ------------------- Originality: The proposed algorithm is new. It can be seen as a variant of the previous proposed underdamped Langevin algorithm (Cheng et al. 2017) with better discretization scheme. Quality: The main proofs (of the main result in Theorem 3) are complete and correct: I checked the proof of lemma 2 and proof of Theorem 3 in the Appendix. There is no simulation analysis. It would be great to see whether one achieves this epsilon^{1/3) improvement via randomized midpoint method in simulation analysis. Clarity: The organization of the main paper is good. However, the writing can be improved. For example, 1. Make a distinction between Markov process and Markov chain algorithm which is a discretized version of the corresponding Markov process. For instance, line 99 directly uses LD both the Langevin diffusion and Langevin algorithm. 2. Introduce notations before using. For instance, line 109, when ULD was introduced, it was not clear what variable u mean here. Also, line 169, it was not clear what v_n(h) is. 3. In Algorithms 1, W_1, W_2, W_3 were not only introduced in Appendix, making the reading of the main paper difficult. 4. The statement of the Theorem 3 is awkward: what does it mean "Algorithm 1 can find a random point X"? Maybe a statement on the distribution at step N is better. Significance: This paper improves the previous rate for log-concave sampling and provides new ideas for designing new sampling algorithms.

[Author Response · NeurIPS 2019]

We sincerely appreciate the time and the efforts the reviewers invested in reading our paper and providing valuable
feedback. We would like to emphasize again the main contribution of our paper. In our paper, we developed a general
framework for simulating stochastic differential equations(SDE). We illustrated the usefulness of our framework on
the sampling problem and obtained a significantly better result than numerous previous results without making any
extra assumptions. We believe that there are many other applications of our framework and that our paper is among
the top accepted papers.

To Reviewer 1: Thanks for the citations and the correction you provided. In our final submission, we will cite [4]
and compare the runtime of our algorithm with [2], which achieves $O\left(\frac{\kappa^{1.5}}{\epsilon} + \kappa^2\right)$ runtime and improves the result
of [1]. We will also correct all the items mentioned in SPECIFIC REMARKS/TYPOS. However, we still prefer using
$O(1)$ instead of explicit numerical constants because we believe numerical constants will distract readers from the key
contributions.

To Reviewer 2: The problem studied in our paper, sampling from log-concave distributions, is an essential tool for
Bayesian inference. It also has many other applications such as volume computation and bandit optimization. (We
mentioned the applications in the first paragraph.) We will make the application of our algorithm more explicit in our
final submission. We will also include an experiment section in our final submission, which shows the performance of
algorithm on real-world datasets. The preliminary results are attached.

To Reviewer 3: We will make a distinction between a Markov process and its discretization in our final submission.
We will make sure that everything is defined properly before used and polish the statement of our theorems. Thanks
for suggesting we evaluate the $\epsilon$ dependence of our paper via experiment. We will include an experiment section in our
final submission which will analyze the performance of our algorithm on real-world datasets. The preliminary results
are attached. The result shows that the bound $\epsilon^{2/3}$ we obtained is in fact tight even for real-world example and is an
improvement over [1].

To All: We attach the preliminary result of our experiments here. In our experiment, we compare the algorithm from
our paper with the one from [1]. In our final submission, we will compare our algorithm with more algorithms. We test
the algorithms on the liver-disorders dataset and the breast-cancer dataset from UCL machine learning repository[3].
For both datasets, we sample from the problem $f(x) = \frac{\lambda}{2}\|x\|^2 + \frac{1}{m}\sum_{i=1}^{m} \log\left(\exp\left(-y_i a_i^T x\right) + 1\right)$ ,where $\lambda$ is the
regularization parameters (We set it to be $10^{-2}$), $y_i$ is the label, $a_i$ is the input and $m$ is the number of inputs. Our
results show that the $\epsilon$ dependence analysis of our algorithm and that of [1] are both tight.

## References

[1] Xiang Cheng, Niladri S Chatterji, Peter L Bartlett, and Michael I Jordan. Underdamped Langevin MCMC: A
non-asymptotic analysis. *arXiv preprint arXiv:1707.03663*, 2017.

[2] Arnak S Dalalyan and Lionel Riou-Durand. On sampling from a log-concave density using kinetic Langevin
diffusions. *arXiv preprint arXiv:1807.09382*, 2018.

[3] Dheeru Dua and Casey Graff. UCI machine learning repository, 2017.

[4] Alain Durmus, Szymon Majewski, and Blazej Miasojedow. Analysis of langevin monte carlo via convex opti-
mization. *Journal of Machine Learning Research*, 20(73):1–46, 2019.


[Meta-Review · NeurIPS 2019]

Many thanks for your submission. We hope the several references provided by reviewers' can help you improve your draft. Reviewers have found the addition of experimental validation useful. Please follow their advice to improve the presentation whenever possible, this is in the interest of everyone attending the conference.